# TPP-SD: Accelerating Transformer Point Process Sampling with Speculative Decoding

**Shukai Gong**[1*]    **Yiyang Fu**[2*]    **Fengyuan Ran**[3*]    **Quyu Kong**[4]    **Feng Zhou**[1†]

[1]Center for Applied Statistics and School of Statistics, Renmin University of China
[2]School of Information, Renmin University of China
[3]School of Cyber Science and Engineering, Wuhan University
[4]Alibaba Group
[1,2]{shukai_gong, fuyiyang2022201505, feng.zhou}@ruc.edu.cn
[3]rfy_Reflow@whu.edu.cn
[4]kongquyu.kqy@alibaba-inc.com

## Abstract

We propose TPP-SD, a novel approach that accelerates Transformer temporal point process (TPP) sampling by adapting speculative decoding (SD) techniques from language models. By identifying the structural similarities between thinning algorithms for TPPs and speculative decoding for language models, we develop an efficient sampling framework that leverages a smaller draft model to generate multiple candidate events, which are then verified by the larger target model in parallel. TPP-SD maintains the same output distribution as autoregressive sampling while achieving significant acceleration. Experiments on both synthetic and real datasets demonstrate that our approach produces samples from identical distributions as standard methods, but with 2-6$\times$ speedup. Our ablation studies analyze the impact of hyperparameters such as draft length and draft model size on sampling efficiency. TPP-SD bridges the gap between powerful Transformer TPP models and the practical need for rapid sequence sampling. Code is publicly available at `https://github.com/GONGSHUKAI/tppsd`.

## 1 Introduction

Temporal point processes (TPPs) [6] are essential stochastic models for describing discrete event occurrence patterns in continuous time. Classical models include the Poisson process [12], which is history-independent, and the Hawkes process [10, 39, 37], which incorporates historical dependencies. Recently, advancements in deep learning have facilitated the integration of deep models with point processes, significantly enhancing their expressive capabilities in capturing history dependence [19]. Notable examples include models based on traditional RNNs [7], LSTMs [17], and Transformers [40, 36, 20]. RNN-type methods often suffer from vanishing or exploding gradients [26], limiting their performance. In contrast, Transformer TPPs have gained popularity for their ability to capture long-term dependencies and support parallel training.

Most current Transformer TPPs' research focuses on enhancing model expressiveness and training methodologies, with limited attention to improving sampling efficiency. Sampling from a learned temporal point process is essential for synthesizing event sequence data, making predictions based on observed history, validating model goodness-of-fit, and gaining insights into complex process dynamics. The standard sampling procedure employs the thinning algorithm [15, 23], which simulates

---

[*]Equal contribution.
[†]Corresponding author.

events from a target process with conditional intensity $\lambda^*(t)$ by iteratively generating candidates $\tilde{t}_{i+1}$ from a simple proposal process (e.g., homogeneous Poisson process with rate $\bar{\lambda}$) and accepting with the probability $\frac{\lambda^*(\tilde{t}_{i+1})}{\bar{\lambda}}$, repeating until acceptance. This is highly inefficient because evaluating the acceptance probability for each proposed candidate requires a forward pass through the Transformer, potentially requiring many forward passes per single accepted event. Despite KV caching [27] reducing inference complexity from quadratic to linear, the large number of parameters in the Transformer architecture makes each forward pass computationally expensive.

Speculative decoding (SD) [5, 14] is a technique designed to accelerate token generation in large language models (LLMs). It generates multiple tokens in parallel, rather than predicting each token sequentially, thereby improving the efficiency of the sampling process. Drawing inspiration from the analogy between next-event sampling in TPPs and next-token prediction in LLMs, we investigate the potential application of SD from LLMs to Transformer TPPs, aiming to enhance sampling efficiency.

Specifically, we make the following key contributions in this work: (1) We identify a strong similarity between the thinning algorithm used in point process sampling and the SD technique in LLMs. (2) We propose TPP-SD, which successfully applies SD to Transformer TPPs, resulting in accelerated sampling. (3) Extensive experiments are conducted on both synthetic and real datasets. The results show that TPP-SD and standard autoregressive sampling produce samples from the same distribution, but TPP-SD achieves a speedup of approximately 2–6×. Ablation studies analyze the impact of different hyperparameters.

## 2 Preliminary Knowledge

In this section, we provide background on TPPs, the thinning algorithm for TPPs, and SD for LLMs.

### 2.1 Temporal Point Processes

TPPs model discrete event sequences in continuous time, denoted as $\mathcal{S} = \{(t_i, k_i)\}_{i=1}^N$ where $(t_i, k_i)$ indicates an event of type $k_i \in \mathcal{K} = \{1, \dots, K\}$ occurring at time $t_i \in (0, T]$, with $0 < t_1 < \cdots < t_N \leq T$ where $N$ denotes the random number of events. TPPs can be defined using different parameterizations. One approach is to specify the conditional distribution of events. We denote $f(t_{i+1}, k_{i+1}|\mathcal{H}_{t_i})$ to be the conditional density function (CDF) of event $(t_{i+1}, k_{i+1})$ given the history of previous events $(t_1, k_1), \dots, (t_i, k_i)$. In this work, $\mathcal{H}_{t^-}$ denotes the history of events up to but excluding time $t$, while $\mathcal{H}_t$ includes whether an event occurs at time $t$. Consequently, the joint distribution of all events can be factorized as:

$$p(\mathcal{S}) = \prod_{i=1}^N f(t_i, k_i|\mathcal{H}_{t_{i-1}})\left(1 - F(T|\mathcal{H}_{t_N})\right),$$

where $F(T|\mathcal{H}_{t_N}) = \int_{t_N}^T f(t|\mathcal{H}_{t_N})\mathrm{d}t$, $f(t|\mathcal{H}_{t_i}) = \sum_{k=1}^K f(t, k|\mathcal{H}_{t_i})$. As is customary, we further decompose $f(t, k|\mathcal{H}_{t_i}) = f(t|\mathcal{H}_{t_i})f(k|\mathcal{H}_{t_i}, t)$. The term $(1 - F(T|\mathcal{H}_{t_N}))$ is the probability that no event occurs in $(t_N, T)$. Another approach is to specify the conditional intensity function (CIF):

$$\lambda^*(t, k)\mathrm{d}t\mathrm{d}k = \frac{f(t, k|\mathcal{H}_{t_i})\mathrm{d}t\mathrm{d}k}{1 - F(t|\mathcal{H}_{t_i})} = \mathbb{E}[N(\mathrm{d}t \times \mathrm{d}k)|\mathcal{H}_{t^-}],$$

where * indicates that the CIF is dependent on the history $\mathcal{H}_{t^-}$. These different parameterizations are equivalent because it can be easily proven that the CIF $\lambda^*(t, k)$ and the CDF $f(t, k|\mathcal{H}_{t_i})$ are one-to-one [38]. It is worth noting that the event timestamps can also be equivalently expressed as inter-event intervals $\tau_i = t_i - t_{i-1}$, $i = 1, \dots, N$ with $t_0 = 0$. Therefore, the CDF of the timestamps can also be equivalently expressed as the CDF of the inter-event intervals $g(\tau|\mathcal{H}_{t_i}) = f(t_i + \tau|\mathcal{H}_{t_i})$. Correspondingly, TPPs have two equivalent forms of log-likelihood expression:

$$\log p(\mathcal{S}) = \sum_{i=1}^N \log \lambda^*(t_i, k_i) - \sum_{k=1}^K \int_0^T \lambda^*(t, k)\mathrm{d}t \tag{1}$$

$$= \sum_{i=1}^N \left[\log g(\tau_i|\mathcal{H}_{t_{i-1}}) + \log f(k_i|\mathcal{H}_{t_{i-1}}, t_i)\right] + \log(1 - G(T|\mathcal{H}_{t_N})), \tag{2}$$

where $G(T|\mathcal{H}_{t_N}) = \int_0^{T-t_N} g(\tau|\mathcal{H}_{t_N})\mathrm{d}\tau$. Eq. (1) is CIF-based, and Eq. (2) is CDF-based.

## 2.2 Thinning Algorithm for TPPs

For illustration purposes, we discuss the sequential simulation of timestamps from a point process using the thinning algorithm. Given the event history $\mathcal{H}_{t_i}$, to sample a timestamp from the original point process model (denoted as OriP) parameterized by CIF $\lambda^*(t)$, we use a proposal point process, typically a homogeneous Poisson process (denoted as PoiP) with intensity $\bar{\lambda} = \sup_{t \in (t_i, \infty)} \lambda^*(t)$, to sample the next candidate timestamp:

$$\tilde{t}_{i+1} \sim \text{PoiP}(\mathcal{H}_{t_i}).$$

Next, the original point process model evaluates the candidate timestamp by calculating the conditional intensity at the given position:

$$\lambda^*(\tilde{t}_{i+1}) = \text{OriP}(\tilde{t}_{i+1}; \mathcal{H}_{t_i}).$$

Then, a rejection sampling step is applied to verify the candidate timestamp. The $\tilde{t}_{i+1}$ is accepted if

$$\epsilon < \frac{\lambda^*(\tilde{t}_{i+1})}{\bar{\lambda}}, \quad \epsilon \sim \text{Uniform}[0, 1].$$

This rejection sampling ensures that the final sequence adheres to the original point process distribution. If the candidate timestamp is rejected, the process is repeated until a candidate is accepted. The accepted timestamp is then added to the event history, forming $\mathcal{H}_{t_{i+1}}$, and the procedure continues until the sequence is fully generated. The efficiency of the thinning algorithm depends heavily on the alignment between the proposal point process and the original point process. A closer alignment results in a higher acceptance rate, improving overall performance.

## 2.3 Speculative Decoding for LLMs

Autoregressive sampling in LLMs is inherently sequential and inefficient. SD addresses this inefficiency by delegating auto-regressive sampling to a smaller language model (draft model, denoted as $\text{LLM}_{\text{dra}}$), which generates multiple candidate tokens sequentially given the context $\mathbf{s}$:

$$(c_1, q_1), \ldots, (c_\gamma, q_\gamma) \sim \text{LLM}_{\text{dra}}(\mathbf{s}),$$

where $\gamma$ is the length of candidate tokens, $c_1, \ldots, c_\gamma$ are the sampled tokens, and $q_1, \ldots, q_\gamma$ are their corresponding probability distributions. Then, the original language model (target model, denoted as $\text{LLM}_{\text{tar}}$) processes these candidate tokens in parallel, producing probability distributions:

$$p_1, \ldots, p_{\gamma+1} = \text{LLM}_{\text{tar}}(c_1, \ldots, c_\gamma; \mathbf{s}),$$

where the extra probability distribution $p_{\gamma+1}$ corresponds to processing the candidate token $c_\gamma$. A sequence of token-level rejection sampling is then applied to verify the candidate tokens. A candidate token $c_i$ is accepted if all previous tokens are accepted and

$$\epsilon_i < \frac{p_i(c_i)}{q_i(c_i)}, \quad \epsilon_i \sim \text{Uniform}[0, 1].$$

This rejection sampling ensures that the final sequence adheres to the distribution of the target model. If $c_i$ is the first rejected token, a replacement can be sampled from an adjusted distribution $\text{norm}(\max(0, p_i - q_i))$. When all candidates are accepted, an additional token is sampled from $p_{\gamma+1}$. The accepted tokens are then added to the context $\mathbf{s}$, and the process repeats until the sequence is complete. The efficiency of SD heavily depends on the alignment between the draft and target models. A closer alignment leads to a higher acceptance rate, improving overall performance.

# 3 Related Works

In this section, we discuss related works on point process sampling. In traditional statistical point processes, the CIF parameterization is generally preferred because the CDF is not a convenient way to specify history-dependent point processes. In contrast, the CIF provides a more convenient way to specify how the present depends on the past. Therefore, sampling methods in the statistical point process field are generally based on the CIF. Inverse method [28] simulates a unit-intensity

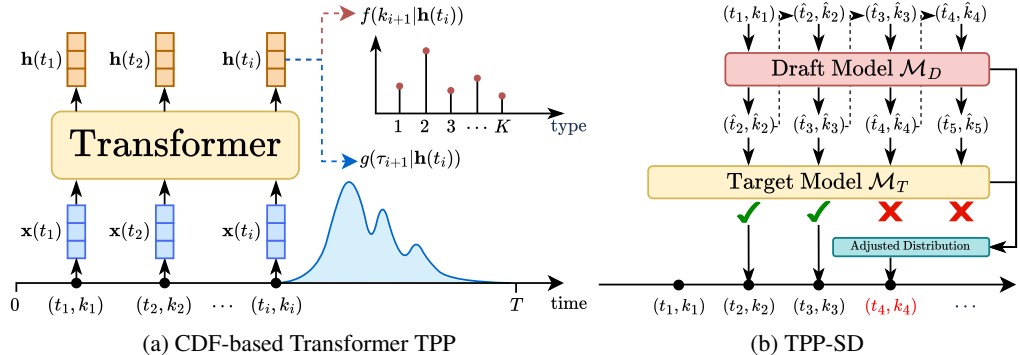

Figure 1: (a) Overall architecture of our proposed CDF-based Transformer TPP. (b) The visualization of our proposed TPP-SD sampling method as elaborated in Section 4.3.

Poisson process and transforms it into the desired point process using the inverse of the integrated CIF, leveraging the time-rescaling theorem [2], and [30] applies the inverse map in parallel to accelerate this process. However, this method is not suitable for complex processes that do not have analytical forms for the inverse of integrated CIF. Another more easily implementable sampling method is the thinning algorithm, which was originally proposed for sampling from inhomogeneous Poisson processes [15], and later adapted for history-dependent processes [23, 31]. With the development of the deep neural point process, an increasing number of works have started using the CDF parameterization. For example, [29] used normalizing flows and mixtures of log-normal densities to model the CDF of the inter-event interval. Although this approach was initially proposed for training convenience, another advantage it brings is facilitating sampling. This is because we can easily sample the next event directly from the CDF [24]. Whether based on CIF or CDF, both of the above methods for sampling history-dependent point processes can only be performed sequentially, leading to inefficient sampling. The key difference of this work is the introduction of parallelization in sampling, which accelerates the sampling process.

## 4 Methodology

In this section, we analyze the similarity between the thinning algorithm and the SD technique. Inspired by this insight, we propose one SD method for Transformer TPP models.

### 4.1 Thinning Algorithm v.s. Speculative Decoding

As introduced in Sections 2.2 and 2.3, the thinning algorithm and SD share significant structural similarities despite their different application domains. Both methods adopt a two-stage "propose-verify" framework that uses a draft model to generate candidates and uses the target model for verification. Besides, the efficiency of both methods critically depends on the alignment between the proposal and target distributions.

Despite these similarities, several key differences distinguish these approaches. The thinning algorithm operates strictly in an autoregressive manner, generating and verifying a single candidate timestamp per propose-verify iteration. In contrast, SD can propose and validate multiple tokens per propose-verify iteration, resulting in accelerated sampling. Additionally, while the thinning algorithm may fail to produce a valid timestamp per propose-verify iteration, SD guarantees the generation of at least one valid token per propose-verify iteration.

Motivated by these observations—especially the algorithmic similarities and the limitations of the thinning algorithm—we propose TPP-SD: a novel CDF-based sampling method for Transformer TPPs that applies the principles of SD to enhance sampling efficiency.

### 4.2 CDF-based Transformer TPPs

To accelerate TPP sampling using SD, we first need to design a CDF-based Transformer TPP model, which is composed of two parts. A similar model design has also been adopted by [24].

**Encoder.** We use Transformer backbones to model the historical dependencies in the event sequence. Specifically, given an observed realization of the point process $\mathcal{S} = \{(t_i, k_i)\}_{i=1}^{N}$, we apply the temporal encoding method from [40] to encode each timestamp $t_i$ into a fixed-length vector $\mathbf{z}(t_i) \in \mathbb{R}^D$. In addition to the temporal encoding, we use an embedding matrix $\mathbf{W} \in \mathbb{R}^{K \times D}$, where the $D$-dimensional embedding for event type $k_i$ is given by $\mathbf{W}^\top \mathbf{k}_i$, with $\mathbf{k}_i \in \mathbb{R}^K$ being the one-hot encoding of $k_i$. Thus, the embedding for the observed realization is given by

$$\mathbf{X} = f(\mathbf{KW}, \mathbf{Z}) \in \mathbb{R}^{N \times D},$$

where $\mathbf{K} = [\mathbf{k}_1, \quad \ldots, \quad \mathbf{k}_N]^\top \in \mathbb{R}^{N \times K}$ and $\mathbf{Z} = [\mathbf{z}_1, \quad \ldots, \quad \mathbf{z}_N]^\top \in \mathbb{R}^{N \times D}$ are the collections of event types and timestamp embeddings of the observed realization, and $f(\cdot, \cdot)$ is a fusion function where we use a summation operation. A Transformer is then applied to obtain the history embedding matrix $\mathbf{H} = T_{\boldsymbol{\theta}}(\mathbf{X}) \in \mathbb{R}^{N \times D}$. Each row of the history embedding matrix $\mathbf{h}^\top(t_i) = \mathbf{H}(i, :) \in \mathbb{R}^D$ corresponds to the history information up to timestamp $t_i$. The choice of encoder $T_{\boldsymbol{\theta}}$ can be any Transformer TPP, such as THP [40], SAHP [36], or AttNHP [18]. The necessity of using a Transformer encoder lies in its ability to enable parallel computation, which is essential for SD.

**Decoder.** Inspired by [24, 29], to enhance the flexibility of the model, we use a mixture of log-normal distributions to model the CDF of the next timestamp $g(\tau_{i+1}|\mathbf{h}(t_i))$, since the inter-event intervals are positive, i.e.,

$$g_{\boldsymbol{\theta}}(\tau_{i+1}|\mathbf{h}(t_i)) = \sum_{m=1}^{M} w_{im} \frac{1}{\tau \sqrt{2\pi} \sigma_{im}} \exp\left(-\frac{(\log \tau_{i+1} - \mu_{im})^2}{2\sigma_{im}^2}\right).$$

Here, $g_{\boldsymbol{\theta}}(\tau_{i+1}|\mathbf{h}(t_i))$ is referred to as the decoder because it takes the history embedding $\mathbf{h}(t_i)$ as input and decodes the inter-event intervals. The mixture weights $\mathbf{w}_i = [w_{i1}, \ldots, w_{iM}]^\top$, the mixture means $\boldsymbol{\mu}_i = [\mu_{i1}, \ldots, \mu_{iM}]^\top$, and the mixture standard deviations $\boldsymbol{\sigma}_i = [\sigma_{i1}, \ldots, \sigma_{iM}]^\top$ are obtained by mapping $\mathbf{h}(t_i)$. Specifically, $\mathbf{h}(t_i) \in \mathbb{R}^D$ is projected to $\mathbf{e}_i = \mathbf{E}\mathbf{h}(t_i) \in \mathbb{R}^{3D}$ with $\mathbf{E} \in \mathbb{R}^{3D \times D}$, and then sliced into three equal-length parts $\mathbf{e}_i = \left[\mathbf{e}_1^{i\top}, \mathbf{e}_2^{i\top}, \mathbf{e}_3^{i\top}\right]^\top$. Subsequently, $\mathbf{e}_1^i, \mathbf{e}_2^i, \mathbf{e}_3^i$ are mapped to $\mathbf{w}_i, \boldsymbol{\mu}_i$, and $\boldsymbol{\sigma}_i$ as follows:

$$\mathbf{w}_i = \text{softmax}(\mathbf{V_w} \mathbf{e}_1^i + \mathbf{b_w}), \quad \boldsymbol{\mu}_i = \mathbf{V_\mu} \mathbf{e}_2^i + \mathbf{b_\mu}, \quad \boldsymbol{\sigma}_i = \exp(\mathbf{V_\sigma} \mathbf{e}_3^i + \mathbf{b_\sigma}),$$

where $\mathbf{V_w}, \mathbf{V_\mu}, \mathbf{V_\sigma} \in \mathbb{R}^{M \times D}$, and $\mathbf{b_w}, \mathbf{b_\mu}, \mathbf{b_\sigma} \in \mathbb{R}^M$ are learnable parameters. The CDF of the next event type $f_{\boldsymbol{\theta}}(k_{i+1}|\mathbf{h}(t_i))$ is modeled as a categorical distribution:

$$f_{\boldsymbol{\theta}}(k_{i+1}|\mathbf{h}(t_i)) = \text{softmax}\left(\mathbf{V_k}^{(2)} \tanh(\mathbf{V_k}^{(1)} \mathbf{h}(t_i) + \mathbf{b_k}^{(1)}) + \mathbf{b_k}^{(2)}\right).$$

Our proposed CDF-based Transformer TPP model is illustrated in Fig. 1a. This model can be formally represented as $\mathcal{M} = \{\mathcal{E}, g(\tau|\cdot), f(k|\cdot)\}$, where $\mathcal{E}$ corresponds to the encoder and $g(\tau|\cdot), f(k|\cdot)$ correspond to the decoder. The model is trained by maximizing the log-likelihood in Eq. (2).

**Naïve autoregressive sampling.** The next timestamp is easy to sample from the log-normal mixture distribution $g_{\boldsymbol{\theta}}(\tau_{i+1}|\mathbf{h}(t_i))$ by (see Appendix A.1 for proof)

$$\mathbf{z}_i \sim \text{Categorical}(\mathbf{w}_i), \quad \epsilon \sim \mathcal{N}(0, 1), \quad \hat{\tau}_{i+1} = \exp(\boldsymbol{\mu}_i^\top \mathbf{z}_i + \epsilon \cdot \boldsymbol{\sigma}_i^\top \mathbf{z}_i),$$

so the next timestamp is $\hat{t}_{i+1} = t_i + \hat{\tau}_{i+1}$. The next event type is sampled by $\hat{k}_{i+1} \sim f_{\boldsymbol{\theta}}(k_{i+1}|\mathbf{h}(t_i))$. The new event $(\hat{t}_{i+1}, \hat{k}_{i+1})$ is appended to the history and the sampling procedure is repeated until we reach the predetermined end time. However, naïve autoregressive sampling requires a forward pass of the Transformer encoder for each event, which becomes inefficient as the number of parameters in the Transformer backbone increases.

### 4.3 TPP-SD

Suppose we have a trained target Transformer TPP model $\mathcal{M}_T$ with a large number of parameters, from which we wish to sample events, and a trained draft Transformer TPP model $\mathcal{M}_D$ with fewer parameters, which we use to approximate $\mathcal{M}_T$. The procedure of TPP-SD consists of three steps, as shown in Fig. 1b and Algorithm 1.

**Drafting.** Sample $\gamma$ candidate events $\{(\hat{t}_{i+1}, \hat{k}_{i+1}), \ldots, (\hat{t}_{i+\gamma}, \hat{k}_{i+\gamma})\}$ autoregressively from $\mathcal{M}_D$. Meanwhile, we record the interval CDF $g_D(\hat{\tau}_{i+l}|\cdot)$ and type CDF $f_D(\hat{k}_{i+l}|\cdot)$ for all candidate events.

**Verification.** Run $\mathcal{M}_T$ in parallel to compute $g_T(\hat{\tau}_{i+l}|\cdot)$ and $f_T(\hat{k}_{i+l}|\cdot)$ for all candidate events. Then, calculate the acceptance rates $\frac{g_T(\hat{\tau}_{i+l}|\cdot)}{g_D(\hat{\tau}_{i+l}|\cdot)}$ and $\frac{f_T(\hat{k}_{i+l}|\cdot)}{f_D(\hat{k}_{i+l}|\cdot)}$ for all candidate events. A candidate inter-event interval $\hat{\tau}_{i+l}$ is accepted if all previous events are accepted and $\epsilon_l^\tau < \frac{g_T(\hat{\tau}_{i+l}|\cdot)}{g_D(\hat{\tau}_{i+l}|\cdot)}, \epsilon_l^\tau \sim$ Uniform$[0,1]$. A candidate event type $\hat{k}_{i+l}$ is accepted if all previous events and $\hat{\tau}_{i+l}$ are accepted, and $\epsilon_l^k < \frac{f_T(\hat{k}_{i+l}|\cdot)}{f_D(\hat{k}_{i+l}|\cdot)}, \epsilon_l^k \sim$ Uniform$[0,1]$.

**Sampling from adjusted distribution.** Once a candidate event interval $\hat{\tau}_{i+l}$ or type $\hat{k}_{i+l}$ is rejected, all subsequent candidate events are automatically discarded and a replacement $\hat{\tau}_{i+l}$ or $\hat{k}_{i+l}$ is sampled from an adjusted distribution defined as below:

$$g'(\tau_{i+l}|\cdot) = \text{norm}(\max(0, g_T(\tau_{i+l}|\cdot) - g_D(\tau_{i+l}|\cdot))), \tag{3}$$

$$f'(k_{i+l}|\cdot) = \text{norm}(\max(0, f_T(k_{i+l}|\cdot) - f_D(k_{i+l}|\cdot))), \tag{4}$$

where norm$(\cdot)$ denotes the normalization. The correctness and necessity of sampling from the adjusted distribution have been established in the original speculative decoding work for LLMs [14]. In this work, we extend the proof to the TPP domain, as presented in Appendix A.2.

Eq. (4) is a discrete distribution, and its normalization is relatively easy to implement, but Eq. (3) is a continuous distribution, and normalizing it is much more difficult because we need to compute the normalizing constant $\int \max(0, g_T(\tau_{i+l}|\cdot) - g_D(\tau_{i+l}|\cdot))d\tau_{i+l}$. This is also the main difference between TPP-SD and LLM-SD, as continuous distributions are not involved in the LLM application. Inspired by [32], we employ an acceptance-rejection sampling scheme [4] to sample from $g'(\tau_{i+l}|\cdot)$, as outlined in Theorem 1 (refer to Appendix A.3 for the proof).

**Theorem 1.** *For a sample $\hat{\tau}_{i+l} \sim g_T(\tau_{i+l}|\cdot)$, define the acceptance threshold as*

$$\alpha = \frac{\max(0, g_T(\hat{\tau}_{i+l}|\cdot) - g_D(\hat{\tau}_{i+l}|\cdot))}{g_T(\hat{\tau}_{i+l}|\cdot)}.$$

*Accept $\hat{\tau}_{i+l}$ if $\epsilon < \alpha$, where $\epsilon \sim$ Uniform$(0,1)$. This acceptance-rejection procedure generates samples from the adjusted distribution $g'(\tau_{i+l}|\cdot)$ defined in Eq. (3).*

## 5 Experiments

In this section, we compare TPP-SD with autoregressive sampling (abbreviated as AR sampling) on both synthetic and real datasets. We verify that both methods produce samples from the same underlying distribution, and demonstrate that TPP-SD significantly improves sampling efficiency.

### 5.1 Evaluation of Sampling Quality and Speed

Unlike point process fitting, which focuses on the accuracy of predicted event types or timestamp deviations, point process sampling emphasizes whether the samples generated by TPP-SD and AR sampling follow the same distribution (i.e., sampling quality), as well as the efficiency of the sampling process (i.e., sampling speed). Therefore, we define the following evaluation metrics.

**Likelihood Discrepancy (Synthetic and Real).** For synthetic data, where the ground truth is known, we generate samples using both AR sampling and TPP-SD, and measure the discrepancy between the ground-truth likelihood (Eq. (1)) and the model likelihood of the generated samples (Eq. (2)). Specifically, we compute $\Delta\mathcal{L}_{\text{ar}}^{\text{syn}} = |\mathcal{L}_{\text{gt}} - \mathcal{L}_{\text{ar}}|$ for AR sampling and $\Delta\mathcal{L}_{\text{sd}}^{\text{syn}} = |\mathcal{L}_{\text{gt}} - \mathcal{L}_{\text{sd}}|$ for TPP-SD. For real data, where the ground truth is unknown, we measure the discrepancy between the AR sampling likelihood and the TPP-SD likelihood, $\Delta\mathcal{L}^{\text{real}} = |\mathcal{L}_{\text{ar}} - \mathcal{L}_{\text{sd}}|$. Ideally, samples generated by TPP-SD and AR sampling should follow the same distribution—i.e., the ground-truth distribution. A lower likelihood discrepancy indicates higher sample quality.

Table 1: Performance of TPP-SD with draft length $\gamma = 10$ against AR sampling across synthetic datasets and Transformer encoders. We conduct all experiments using three random seeds and report the mean for each metric. For all metrics, the best performance is highlighted in **bold**.

| Dataset | | Poisson | | | Hawkes | | | Multi-Hawkes | | |
|---|---|---|---|---|---|---|---|---|---|---|
| Encoder Type | | THP | SAHP | AttNHP | THP | SAHP | AttNHP | THP | SAHP | AttNHP |
| $\Delta\mathcal{L}^{\text{syn}}$ ($\downarrow$) | AR Sampling | 0.542 | **0.012** | **1.879** | 0.753 | 0.884 | **0.220** | **0.022** | 0.146 | 0.334 |
| | TPP-SD | **0.349** | 0.204 | 1.952 | **0.276** | **0.630** | 0.722 | 0.321 | **0.070** | **0.199** |
| $D_{\text{KS}}$ ($\downarrow$) | AR Sampling | 0.038 | **0.033** | 0.076 | 0.044 | 0.031 | 0.029 | 0.069 | **0.055** | 0.065 |
| | TPP-SD | **0.036** | 0.050 | **0.068** | **0.043** | **0.028** | **0.027** | **0.053** | 0.080 | **0.045** |
| Wall-time $T$ ($\downarrow$) | AR Sampling | 3.477 | 2.680 | 12.103 | 5.147 | 2.747 | 20.503 | 4.007 | 2.490 | 12.403 |
| | TPP-SD | **1.647** | **2.077** | **4.063** | **2.547** | **1.863** | **3.567** | **1.893** | **1.647** | **2.770** |
| Speedup Ratio $S_{\text{AR/SD}}$ ($\uparrow$) | | 2.110 | 1.290 | 2.967 | 2.113 | 1.513 | 5.743 | 2.117 | 1.277 | 4.467 |

**Kolmogorov-Smirnov Statistic (Synthetic Only).** For synthetic data where the ground truth is known, the time-rescaling theorem [2] states that for event times $\{t_i\}_{i=1}^{n}$ generated by a point process with CIF $\lambda^*(t)$, if the model is correctly specified, the transformed inter-event intervals $z_i = \int_{t_{i-1}}^{t_i} \lambda^*(\tau)\, d\tau$ are i.i.d. samples from Exponential(1). Therefore, we apply the ground-truth CIF to transform the samples generated by AR sampling and TPP-SD, and then compute the KS statistic $D_{\text{KS}}$ to assess the conformity of generated samples to the ground-truth distribution. A smaller $D_{\text{KS}}$ indicates higher sample quality. The detailed computation of $D_{\text{KS}}$ is provided in Appendix A.4.

**Wasserstein Distance (Real Only).** For real data where the ground truth is unknown, the KS statistic cannot be computed. Instead, we assess the sampling consistency between AR sampling and TPP-SD using the Wasserstein distance $D_{\text{WS}}$. Specifically, using the first $M$ events as history, we perform $N$ independent repetitions of sampling $(M + 1)$-th event, yielding $\{(t_i^{\text{AR}}, k_i^{\text{AR}})\}_{i=1}^{N}$ from AR sampling and $\{(t_i^{\text{SD}}, k_i^{\text{SD}})\}_{i=1}^{N}$ from TPP-SD. For the temporal distribution, we compute $D_{\text{WS}}^t$, the 1-Wasserstein distance between the empirical distributions of $\{t_i^{\text{AR}}\}_{i=1}^{N}$ and $\{t_i^{\text{SD}}\}_{i=1}^{N}$, using `ot.wasserstein_1d` from the POT library [9]. For the event type distribution, we compute $D_{\text{WS}}^k$, the earth mover's distance between the empirical distributions of $\{k_i^{\text{AR}}\}_{i=1}^{N}$ and $\{k_i^{\text{SD}}\}_{i=1}^{N}$, using `ot.emd2` from the same library. A smaller $D_{\text{WS}}$ indicates higher sample quality.

**Speedup Ratio (Synthetic and Real).** Sampling efficiency is central to this work. We quantify the acceleration gain by computing the ratio of execution wall times between AR sampling and TPP-SD, defined as $S_{\text{AR/SD}} = \frac{T_{\text{AR}}}{T_{\text{SD}}}$, where $T_{\text{AR}}$ and $T_{\text{SD}}$ denote the execution wall times of AR sampling and TPP-SD, respectively. A larger $S_{\text{AR/SD}}$ indicates a faster sampling speed.

## 5.2 Experimental Results on Synthetic Data

**Datasets and Setup.** We consider three synthetic datasets: inhomogeneous Poisson, univariate Hawkes, and multivariate Hawkes processes, each with 1000 sequences within the time window $[0, 100]$. For each dataset, we train an 8-head, 20-layer target model and a 1-head, 1-layer draft model. We compare two sampling approaches: AR sampling using only the target model versus TPP-SD which combines both target and draft models as elaborated in Section 4.3. Details on data simulation procedures, data splitting, and experimental settings are provided in Appendices B.1 and C.3.1.

**Results.** As shown in Table 1, across all three synthetic datasets and three encoder architectures, TPP-SD consistently demonstrates high-fidelity sampling that closely matches the performance of AR sampling. Specifically, both TPP-SD and AR sampling exhibit low likelihood discrepancies and near-zero KS statistics relative to the ground truth, indicating excellent alignment with the target distribution. Furthermore, as shown in Fig. 2, the KS plots show that the samples generated by both methods consistently fall within the 95% confidence bands.

TPP-SD consistently achieves faster execution times and $1.3$–$5.7\times$ speedup across all datasets and encoder architectures. These results demonstrate that TPP-SD maintains the same high sampling fidelity as AR sampling while providing significant improvements in sampling efficiency.

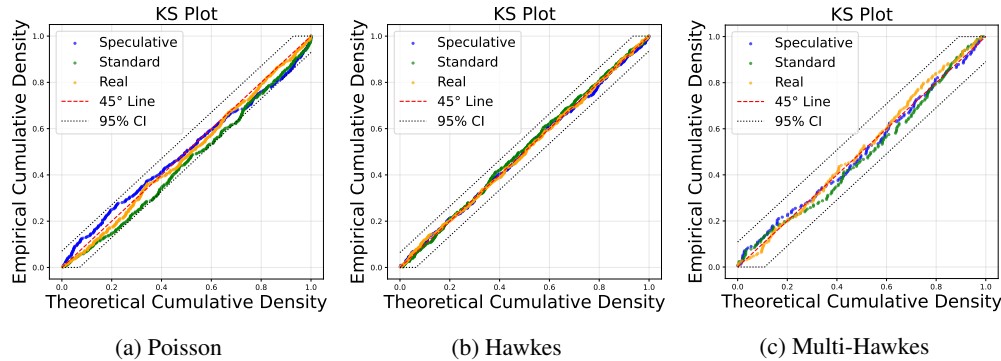

|   | (a) Poisson | (b) Hawkes | (c) Multi-Hawkes |
|---|---|---|---|

Figure 2: KS plots for (a) Poisson, (b) Hawkes, and (c) Multi-Hawkes datasets. We use AttNHP as encoder, and blue, green, and orange points represent samples from TPP-SD ($\gamma = 10$), AR sampling, and ground truth, respectively. Black dotted lines show 95% KS confidence bands.

Table 2: Performance of TPP-SD with draft length $\gamma = 10$ against AR sampling across real datasets and Transformer encoders. We conduct all experiments using three random seeds and report the mean for each metric. For all metrics, the best performance is highlighted in **bold**.

| Dataset | | Taobao | | | Amazon | | | Taxi | | | StackOverflow | | |
|---|---|---|---|---|---|---|---|---|---|---|---|---|---|---|
| Encoder Type | | THP | SAHP | AttNHP | THP | SAHP | AttNHP | THP | SAHP | AttNHP | THP | SAHP | AttNHP |
| $\Delta\mathcal{L}^{\text{real}}$ ($\downarrow$) | AR Sampling | 0.446 | **0.148** | **0.629** | **0.056** | 0.099 | **0.118** | 1.4411 | 0.563 | 0.859 | 0.587 | **0.340** | 0.985 |
| | TPP-SD | **0.033** | 0.746 | 0.860 | 0.129 | **0.035** | 0.197 | **0.065** | **0.093** | **0.506** | **0.231** | 0.602 | **0.020** |
| $D^t_{\text{WS}}$ ($\downarrow$) | AR Sampling | 0.236 | **0.328** | 0.187 | 0.189 | **0.019** | 0.975 | 0.201 | 0.236 | **0.249** | 0.470 | **0.378** | 0.677 |
| | TPP-SD | **0.076** | 0.493 | **0.116** | **0.078** | 0.146 | **0.464** | **0.082** | **0.036** | 0.331 | **0.391** | 0.518 | **0.614** |
| $D^k_{\text{WS}}$ ($\downarrow$) | AR Sampling | **0.267** | 0.414 | **0.193** | **0.184** | 0.459 | **0.252** | **0.055** | 0.778 | **0.094** | 0.376 | 0.381 | **0.218** |
| | TPP-SD | 0.751 | **0.368** | 0.206 | 0.418 | 1.409 | 0.327 | 0.655 | **0.744** | 0.134 | **0.375** | **0.199** | 0.507 |
| Wall-time $T$ ($\downarrow$) | AR Sampling | 5.890 | 2.460 | 16.256 | 1.023 | 0.900 | 7.657 | 1.157 | 1.183 | 2.573 | 1.353 | 1.423 | 3.217 |
| | TPP-SD | **3.460** | **1.643** | **5.180** | **0.290** | **0.317** | **1.353** | **0.453** | **0.347** | **0.650** | **0.700** | **0.663** | **0.783** |
| Speedup Ratio $S_{\text{AR/SD}}$ ($\uparrow$) | | 1.597 | 1.553 | 3.183 | 3.550 | 2.847 | 5.849 | 2.553 | 3.637 | 4.310 | 1.930 | 2.153 | 4.290 |

In addition, architectural variations are evident: AttNHP achieves the highest acceleration ratios despite having the slowest wall times; SAHP shows the lowest speedup but maintains the fastest wall times; and THP strikes a balance between acceleration ratio and runtime. These results suggest that the underlying architecture plays a significant role in determining the effectiveness of acceleration. A detailed discussion of these differences is provided in Appendix D.4.

## 5.3 Experimental Results on Real Data

**Datasets and Setup.** For real data, we consider four commonly used datasets: **Taobao** [1], **Amazon** [22], **Taxi** [33], and **StackOverflow** [8]. We similarly train an 8-head, 20-layer target model and a 1-head, 1-layer draft model as in Section 5.2, and compare the sampling performance between AR sampling and TPP-SD. Details on data statistics, data splitting, and experimental settings are provided in Appendix C.3.2. It is worth noting that for real data, the ground-truth distribution is unknown. Therefore, we use AR sampling as a reference and quantify how closely TPP-SD approximates it. To establish a meaningful baseline that accounts for the inherent stochasticity of TPP sampling, we compare two independent runs of AR sampling. In theory, the likelihood discrepancy and Wasserstein distances between two such runs should be zero. However, due to stochastic variation, even independent autoregressive runs may exhibit small differences. This self-comparison serves as a baseline for evaluating how well TPP-SD aligns with AR sampling.

**Results.** Table 2 shows TPP-SD consistently delivers high-fidelity sampling across all real datasets and encoder architectures. Similar to the observations on synthetic data, $\Delta\mathcal{L}^{\text{real}}$ remains low, and both $D^t_{\text{WS}}$ and $D^k_{\text{WS}}$ approach zero, indicating strong alignment in both temporal and type distributions. Moreover, TPP-SD achieves 1.6–5.9× faster execution across all settings. These results confirm that TPP-SD matches the sampling fidelity of AR sampling while offering substantial gains in efficiency.

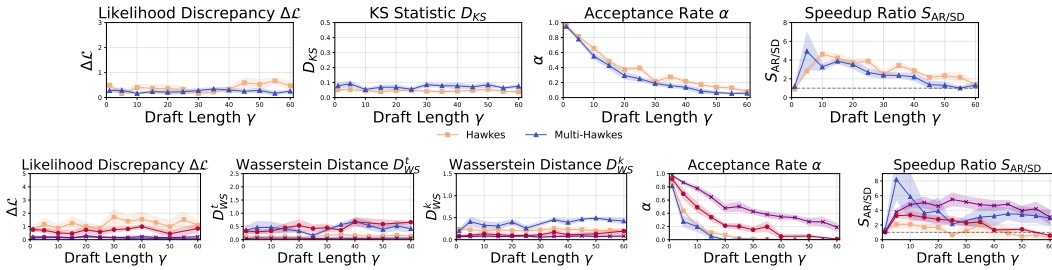

Figure 3: The impact of draft length $\gamma$ on sampling quality measured by likelihood discrepancy ($\Delta\mathcal{L}$) and distance ($D_{\mathrm{KS}}$ or $D_{\mathrm{WS}}$), and on sampling speed measured by speedup ratio $S_{\mathrm{AR/SD}}$. We conduct all experiments using five random seeds and report the mean and the error band for each metric.

Table 3: Performance of TPP-SD with draft length $\gamma = 10$ under different size of draft model. The distance metrics $D_{\mathrm{KS}}$ is used for synthetic datasets, while $D_{\mathrm{WS}}^t$ and $D_{\mathrm{WS}}^k$ are used for real datasets. We conduct all experiments using three random seeds and report the mean for each metric. For all metrics, the best performance is highlighted in **bold**, and the second best is highlighted in underline.

| Dataset | Encoder Type | Draft Model head | Draft Model layer | $\Delta\mathcal{L}$ | $D_{\mathrm{KS}}$ ($\downarrow$) | $D_{\mathrm{WS}}^t$ ($\downarrow$) | $D_{\mathrm{WS}}^k$ ($\downarrow$) | $\alpha$ ($\uparrow$) | $T_{\mathrm{AR}}$ ($\downarrow$) | $T_{\mathrm{SD}}$ ($\downarrow$) | $S_{\mathrm{AR/SD}}$ ($\uparrow$) |
|---|---|---|---|---|---|---|---|---|---|---|---|
| Multi-Hawkes | AttNHP | 1 | 1 | 0.098 | **0.011** | - | - | 0.600 | 12.403 | **2.650** | **4.680** |
| | | 2 | 4 | 0.139 | 0.009 | - | - | 0.710 | 12.403 | 3.003 | 4.130 |
| | | 4 | 6 | **0.227** | 0.004 | - | - | **0.740** | 12.403 | 5.176 | 2.676 |
| Taobao | AttNHP | 1 | 1 | 0.276 | - | **0.080** | 0.197 | 0.220 | 16.256 | **5.727** | **2.838** |
| | | 2 | 4 | **0.174** | - | 0.129 | 0.200 | 0.300 | 16.256 | 6.513 | 2.496 |
| | | 4 | 6 | 0.371 | - | 0.131 | **0.190** | **0.35** | 16.256 | 8.81 | 1.845 |

An interesting observation is that the speedup inversely correlates with event type cardinality. Datasets with higher cardinality ($K = 17$ for Taobao, $K = 22$ for StackOverflow) tend to yield lower speedup compared to those with lower cardinality ($K = 16$ for Amazon, $K = 10$ for Taxi). This is because a larger number of event types increases the probability of divergence between the draft and target models, leading to more rejections during SD. Additionally, the AttNHP encoder consistently achieves greater speedups than THP and SAHP, a trend also observed in the synthetic data experiments.

## 5.4  Ablation Studies

We analyze the sensitivity of two critical hyperparameters, draft length $\gamma$ and draft model size. Throughout this part, we adopt AttNHP encoder, and record the likelihood discrepancy $\Delta\mathcal{L}$ ($\Delta\mathcal{L}_{\mathrm{sd}}^{\mathrm{syn}}$ for synthetic and $\Delta\mathcal{L}^{\mathrm{real}}$ for real datasets), distance metrics $D$ ($D_{\mathrm{KS}}$ for synthetic and $D_{\mathrm{WS}}$ for real datasets), acceptance rate $\alpha = \frac{\#\,\text{events accepted}}{\#\,\text{events drafted}}$, wall time $T$ and speedup ratio $S_{\mathrm{AR/SD}}$.

**Draft Length.**  To identify the optimal draft length $\gamma$ for TPP-SD, we evaluate performance across datasets within the time window $[0, 100]$ for $\gamma \in [1, 60]$. Fig. 3 shows that variations in $\gamma$ have negligible impacts on the metrics $\Delta\mathcal{L}$ and $D$, confirming the distributional equivalence of TPP-SD and AR sampling. However, acceptance rate $\alpha$ decreases as $\gamma$ increases, and speedup $S_{\mathrm{AR/SD}}$ peaks before declining—falling below $1\times$ for excessively large draft lengths due to the overhead of evaluating rejected drafts. Moderate draft lengths (e.g., $\gamma \approx 5$–$15$) yield the highest speedup across all datasets.

**Draft Model Size.**  We investigate the impact of draft model size on performance by fixing the target model to an 8-head, 20-layer Transformer and evaluating three draft configurations: 1-head-1-layer, 2-head-4-layer, and 4-head-6-layer. Experiments are conducted on the Multi-Hawkes and Taobao datasets. Results show that increasing draft model size preserves sampling quality—as measured by $\Delta\mathcal{L}$ and $D$—and improves the acceptance rate $\alpha$, but reduces the speedup ratio $S_{\mathrm{AR/SD}}$. Notably, the 1-head-1-layer configuration achieves the highest speedup without compromising sample quality.

# 6  Conclusions

In this work, we introduce the original SD framework from the LLM domain into the context of TPP sampling. By identifying structural similarities between the thinning algorithm in TPPs and speculative decoding in LLMs, we develop an efficient framework that employs a lightweight draft model to propose candidate events for verification by the target model. TPP-SD significantly improves sampling efficiency by $2$–$6\times$ while preserving distributional consistency with AR sampling, thus bridging the gap between the expressive power of Transformer TPPs and the need for efficient sequence generation in practical applications. As future work, we plan to further optimize TPP-SD by incorporating advancements in SD. Specifically, we aim to integrate the draft mechanism into the target model [3], or perform speculative decoding at the feature level rather than the event level [16].

## Acknowledgments

This work was supported by the NSFC Project (No.62576346), the MOE Project of Key Research Institute of Humanities and Social Sciences (22JJD110001), the fundamental research funds for the central universities, and the research funds of Renmin University of China (24XNKJ13), and Beijing Advanced Innovation Center for Future Blockchain and Privacy Computing.

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

## A  Proof of Mathematical Theorems

### A.1  Sampling from a Log-Normal Mixture Model

To sample a random variable $\tau$ from a log-normal mixture distribution where $w_m$ are the mixture weights, $\mu_m$ are the location parameters and $\sigma_m$ are the scale parameters, $M$ is the number of mixture components

$$p(\tau) = \sum_{m=1}^{M} w_m \cdot \frac{1}{\tau\sqrt{2\pi\sigma_m^2}} \exp\left(-\frac{(\log\tau - \mu_m)^2}{2\sigma_m^2}\right), \tag{5}$$

we only need to do the following procedure

$$z \sim \text{Categorical}(w_1, \ldots, w_M), \ \epsilon \in \mathcal{N}(0,1), \ \tau = \exp(\mu_z + \sigma_z \cdot \epsilon), \tag{6}$$

where $z$ is a single integer $z \in \{1, \ldots, M\}$. By $z \sim \text{Categorical}(w_1, \ldots, w_M)$, we choose the $z-$th mode of the mixture of log-normal with location and scale parameter $(\mu_z, \sigma_z)$. Then by using the change-of-variable formula

$$p(\tau) = p(\epsilon) \left|\frac{d\epsilon}{d\tau}\right| = \frac{1}{\sqrt{2\pi}} \cdot \exp\left(-\frac{(\log\tau - \mu_z)^2}{2\sigma_z^2}\right) \frac{1}{\sigma_z\tau} = \frac{1}{\tau\sqrt{2\pi}\sigma_z} \exp\left(-\frac{(\log\tau - \mu_z)^2}{2\sigma_z^2}\right), \tag{7}$$

which is exactly the PDF of $\mathcal{LN}(\mu_z, \sigma_z)$.

### A.2  Correctness of Sampling from the Adjusted Distribution

Denote $\beta_\tau$ as the acceptance probability of candidate event interval in SD, i.e.

$$\beta_\tau = \mathbb{E}_{\tau \sim g_D(\tau_{i+l}|\cdot)} \min\left(1, \frac{g_T(\tau_{i+l}|\cdot)}{g_D(\tau_{i+l}|\cdot)}\right) = \int \min\left(g_T(\tau|\cdot), g_D(\tau|\cdot)\right) d\tau. \tag{8}$$

Note that

$$g'(\tau_{i+l}|\cdot) = \frac{\max(0, g_T(\tau_{i+l}|\cdot) - g_D(\tau_{i+l}|\cdot)}{\int \max\left(0, g_T(\tau_{i+l}|\cdot) - g_D(\tau_{i+l}|\cdot)\right) d\tau_{i+l}} = \frac{g_T(\tau_{i+l}|\cdot) - \min(g_D(\tau_{i+l}|\cdot), g_T(\tau_{i+l}|\cdot))}{1 - \beta_\tau}. \tag{9}$$

Denote $P(\text{accept}, \tau')$ as the probability that the sample $\tau'$ drawn by the draft model is accepted by the target model, and $P(\text{reject}, \tau')$ as the probability that $\tau'$ is rejected and subsequently resampled from the adjusted distribution defined in Eq. (3). By the law of total probability, the distribution of a sample $\tau'$ drawn by SD can be calculated as

$$p(\tau') = P(\text{accept}, \tau') + P(\text{reject}, \tau') \tag{10}$$

$$= \min\left(1, \frac{g_T(\tau'|\cdot)}{g_D(\tau'|\cdot)}\right) g_D(\tau'|\cdot) + (1 - \beta_\tau)g'(\tau'|\cdot) \tag{11}$$

$$= \min\left(1, \frac{g_T(\tau'|\cdot)}{g_D(\tau'|\cdot)}\right) g_D(\tau'|\cdot) + (g_T(\tau'|\cdot) - \min(g_D(\tau'|\cdot), g_T(\tau'|\cdot))) \tag{12}$$

$$= g_T(\tau'|\cdot). \tag{13}$$

which justifies the correctness of sampling from an adjusted distribution. The derivation also shows that sampling $\tau'$ directly from the target distribution $g_T(\tau_{i+L}|\cdot)$ instead of the adjusted distribution $g'(\tau_{i+l}|\cdot)$ after rejection would result in $p(\tau') \neq g_T(\tau'|\cdot)$, leading to incorrect sampling. Similarly, denote the acceptance probability of candidate event type in SD as $\beta_k = \mathbb{E}_{k \sim f_D(k_{i+l}|\cdot)} \min\left(1, \frac{f_T(k_{i+l}|\cdot)}{f_D(k_{i+l}|\cdot)}\right) = \sum_k \min\left(f_T(k|\cdot), f_D(k|\cdot)\right)$, we can also derive the correctness of sampling $k' \sim f'(k_{i+l}|\cdot)$:

$$p(k') = P(\text{accept}, k') + P(\text{reject}, k') \tag{14}$$

$$= \min\left(1, \frac{f_T(k'|\cdot)}{f_D(k'|\cdot)}\right) f_D(k'|\cdot) + (1 - \beta_k)f'(k'|\cdot) \tag{15}$$

$$= \min\left(1, \frac{f_T(k'|\cdot)}{f_D(k'|\cdot)}\right) f_D(k'|\cdot) + (f_T(k'|\cdot) - \min(f_D(k'|\cdot), f_T(k'|\cdot))) \tag{16}$$

$$= f_T(k'|\cdot). \tag{17}$$

## A.3 Proof of Theorem 1

We refer to [32] for the proof in this subsection. We begin by recalling the principle and demonstrating the correctness of the standard acceptance-rejection sampling [4], which aims to draw samples from a target distribution $p(\tau)$ using a proposal distribution $q(\tau)$ such that $p(\tau) \leq Mq(\tau)$, $\forall \tau \in \mathcal{T}$, where $\mathcal{T}$ is the support of $p(\tau)$ and $q(\tau)$, and $M \geq 1$ is a finite constant. The procedure is as follows: (1) Sample a candidate $\hat{\tau} \sim q(\tau)$. (2) Sample a uniform random variable $\epsilon \sim \text{Uniform}(0, 1)$. (3) Accept the candidate $\hat{\tau}$ if $\epsilon < \frac{p(\hat{\tau})}{Mq(\hat{\tau})}$; otherwise, reject it and return to step (1). To prove that the accepted samples follow the distribution $p(\tau)$, let $\mathcal{A}$ be any measurable set within the support $\mathcal{T}$. The probability that an accepted sample $\hat{\tau}$ falls into $\mathcal{A}$ is

$$\mathbb{P}(\hat{\tau} \in \mathcal{A} | \hat{\tau} \text{ is accepted}) = \frac{\mathbb{P}(\hat{\tau} \in \mathcal{A}, \hat{\tau} \text{ is accepted})}{\mathbb{P}(\hat{\tau} \text{ is accepted})} = \frac{\mathbb{P}(\hat{\tau} \in \mathcal{A}, \hat{\tau} \text{ is accepted})}{\mathbb{P}(\hat{\tau} \in \mathcal{T}, \hat{\tau} \text{ is accepted})}. \tag{18}$$

Note that

$$\mathbb{P}(\hat{\tau} \in \mathcal{A}, \hat{\tau} \text{ is accepted}) = \int_{\mathcal{T}} \int_0^1 \mathbb{I}(\hat{\tau} \in \mathcal{A})\mathbb{I}\left(\epsilon < \frac{p(\hat{\tau})}{Mq(\hat{\tau})}\right) p(\hat{\tau})\mathrm{d}\epsilon\mathrm{d}\hat{\tau} \tag{19}$$

$$= \frac{1}{M} \int_{\mathcal{T}} \mathbb{I}(\hat{\tau} \in \mathcal{A})\frac{p(\hat{\tau})}{q(\hat{\tau})}q(\hat{\tau})\mathrm{d}\hat{\tau} = \frac{1}{M} \int_{\mathcal{A}} p(\hat{\tau})\mathrm{d}\hat{\tau}. \tag{20}$$

Therefore,

$$\mathbb{P}(\hat{\tau} \in \mathcal{A} | \hat{\tau} \text{ is accepted}) = \frac{\int_{\mathcal{A}} p(\hat{\tau})\mathrm{d}\hat{\tau}/M}{\int_{\mathcal{T}} p(\hat{\tau})\mathrm{d}\hat{\tau}/M} = \int_{\mathcal{A}} p(\hat{\tau})\mathrm{d}\hat{\tau}, \tag{21}$$

which justifies the correctness of acceptance-rejection sampling. Now we apply this acceptance-rejection sampling framework to the specific scenario described in the theorem. The adjusted distribution $g'(\tau_{i+l}|\cdot)$ defined in Eq. (3) corresponds to the target distribution $p(\tau)$, and $g_T(\tau_{i+l}|\cdot)$ corresponds to the proposal distribution $q(\tau)$. We need to find the constant $M$ such that for any sampled $\hat{\tau}_{i+L}$, $g'(\hat{\tau}_{i+l}|\cdot) \leq Mg_T(\hat{\tau}_{i+l}|\cdot)$. Denote $Z = \int \max(0, g_T(\hat{\tau}_{i+l}|\cdot) - g_D(\hat{\tau}_{i+l}|\cdot))\,\mathrm{d}\hat{\tau}_{i+l}$, note that

$$g'(\hat{\tau}_{i+l}|\cdot) = \frac{\max(0, g_T(\hat{\tau}_{i+l}|\cdot) - g_D(\hat{\tau}_{i+l}|\cdot))}{Z} \leq \frac{g_T(\hat{\tau}_{i+l}|\cdot) - g_D(\hat{\tau}_{i+l}|\cdot)}{Z} \leq \frac{g_T(\hat{\tau}_{i+l}|\cdot)}{Z}, \tag{22}$$

This inequality confirms that we can set $M := 1/Z$ to satisfy the requirement of acceptance-rejection sampling. With $M = \frac{1}{Z}$, the acceptance threshold $\alpha$ is computed as

$$\alpha = \frac{g'(\hat{\tau}_{i+l}|\cdot)}{Mg_T(\hat{\tau}_{i+l}|\cdot)} = \frac{\max(0, g_T(\hat{\tau}_{i+l}|\cdot) - g_D(\hat{\tau}_{i+l}|\cdot))/Z}{g_T(\hat{\tau}_{i+l}|\cdot)/Z} = \frac{\max(0, g_T(\hat{\tau}_{i+l}|\cdot) - g_D(\hat{\tau}_{i+l}|\cdot))}{g_T(\hat{\tau}_{i+l}|\cdot)}. \tag{23}$$

which is precisely the acceptance threshold $\alpha$ defined in Theorem 1.

## A.4 Computation of the KS Statistic

We employ the Kolmogorov-Smirnov (KS) statistic to assess whether sampled realizations conform to ground truth distributions in synthetic point processes. For $n$ i.i.d. ordered observations $\{X_i\}_{i=1}^n$ from a distribution $F(x)$, the KS statistic is defined as $D_{\text{KS}} = \sup_x |F_n(x) - F(x)|$, where $F_n(x)$ is the empirical cumulative distribution of $\{X_i\}_{i=1}^n$. The leverage of KS statistic is guaranteed by Theorem 2.

**Theorem 2** (Time Rescaling Theorem [21, 25]). *Given a point process with CIF $\lambda(t|\mathcal{H}_t)$ and with occurrence times $0 < t_1 < \cdots < t_n \leq T$, define*

$$z_1 = \int_0^{t_1} \lambda(t|\mathcal{H}_t)\mathrm{d}t, \; z_i = \int_{t_{i-1}}^{t_i} \lambda(t|\mathcal{H}_t)\mathrm{d}t, \; i = 2, \ldots, n. \tag{24}$$

*Then $\{z_i\}_{i=1}^n$ are independent exponential random variables with rate parameter 1.*

By transforming sampled timestamps $\{t_i\}_{i=1}^n$ using the ground truth intensity function, we obtain rescaled intervals $\{z_i\}_{i=1}^n$ whose empirical distribution $F_n(x)$ should approximate $F(x) = 1 - e^{-x}$, $x > 0$ if sampling is correct. Low $D_{\text{KS}}$ values indicate high conformity to the ground truth distribution.

# B  Dataset Statistics

## B.1  Synthetic Dataset

We consider three sets of synthetic data, where each dataset contains 1000 sequences and is split into 80%/10%/10% for training/validation/testing. The data are simulated using the traditional thinning algorithm [15, 23].

**Poisson**  is simulated from an inhomogeneous Poisson process with an intensity function $\lambda(t) = A\left(b + \sin\left(\omega\pi t\right)\right)$ with $A = 5, b = 1, \omega = \frac{1}{50}$ and time span $T = 100$.

**Hawkes**  is simulated from a single-variate Hawkes process $\lambda(t) = \mu + \sum_{t_i < t} \alpha \exp(-\beta(t - t_i))$ with $\mu = 2.5, \alpha = 1, \beta = 2$ and time span $T = 100$.

**Multi-Hawkes**  is simulated from a $M-$dimensional Hawkes process, where the intensity for the $j$-th dimension is $\lambda_j(t) = \mu_m + \sum_{i=1}^{M} \sum_{t_i^j < t} \alpha_{ij} \exp(-\beta_{ij}(t - t_i^j))$. Here $M = 2, \mu_1 = \mu_2 = 0.4, \alpha_{11} = \alpha_{22} = 1, \alpha_{12} = 0.5, \alpha_{21} = 0.1, \beta_{11} = \beta_{12} = \beta_{21} = \beta_{22} = 2$.

## B.2  Real Dataset

We consider four sets of real data listed below. For all datasets, we maintained the standard training/validation/testing splits established in prior work.

**Taobao [1]**  captures the temporal behavioral patterns of the 2,000 most engaged anonymous users on the Taobao e-commerce platform during a nine-day period from November 25th to December 3rd, 2017. The dataset encompasses $K = 17$ distinct event categories.

**Amazon [22]**  documents the chronological shopping activities of anonymous customers on the Amazon marketplace from January 2008 to October 2018. The dataset features $K = 16$ different event categories that characterize various user interactions with the platform.

**Taxi [33]**  comprises pickup and dropoff incidents for taxis operating across New York City's five boroughs (Manhattan, Brooklyn, Queens, The Bronx, and Staten Island). Each borough-specific pickup or dropoff action is classified as a distinct event type, yielding a total of $K = 10$ event categories.

**StackOverflow [8]**  consists of sequential badge-earning events by users on the prominent question-answering platform StackOverflow. Individual users accumulate various badges over approximately a two-year timeframe, with the dataset containing $K = 22$ unique badge types in total.

# C  Addtional Experiment Details

## C.1  Algorithm for TPP-SD

The pseudo code for the algorithm of TPP-SD is shown in Algorithm 1.

## C.2  Training Details

To implement our proposed CDF-based Transformer TPP model, we modified the codebase from [29]. The original RNN encoder for history aggregation was replaced with Transformer backbones proposed by THP [40], SAHP [36], and AttNHP [18]. Given that these models are CIF-based, we extracted only their Transformer architectures, utilizing implementations available in the `EasyTPP` [34] GitHub repository[3]. In line with [29], we set the history embedding dimension $D = 64$ and the number of mixture components $M = 64$.

---

[3] `https://github.com/ant-research/EasyTemporalPointProcess`

**Algorithm 1: TPP-SD**

---

1 **Inputs:** Initial event $(t_0, k_0)$, end time $T$, number of draft events $\gamma$, target model $\mathcal{M}_T = \{\mathcal{E}_T, g_T(\tau|\cdot), f_T(k|\cdot)\}$, draft model $\mathcal{M}_D = \{\mathcal{E}_D, g_D(\tau|\cdot), f_D(k|\cdot)\}$;

2 **Initialize:** Sampled events $\mathcal{S} = \{(t_0, k_0)\}$, $i \leftarrow 0$;

3 **while** $t_i < T$ **do**

4      **for** $l = 1$ **to** $\gamma$ **do**

5          Sample $\hat{\tau}_{i+l} \sim g_D(\tau|\cdot)$, record $g_D(\hat{\tau}_{i+l}|\cdot)$ and $\hat{t}_{i+l} \leftarrow t_{i+l-1} + \hat{\tau}_{i+l}$;

6          Sample $\hat{k}_{i+l} \sim f_D(k|\cdot)$ and record $f_D(\hat{k}_{i+l}|\cdot)$;

7      Compute $g_T(\hat{\tau}_{i+1}|\cdot), \ldots, g_T(\hat{\tau}_{i+\gamma}|\cdot)$ and $f_T(\hat{k}_{i+1}|\cdot), \ldots, f_T(\hat{k}_{i+\gamma}|\cdot)$ in parallel;

8      $\epsilon_1^\tau, \ldots \epsilon_\gamma^\tau, \epsilon_1^k, \ldots, \epsilon_\gamma^k \sim \text{Uniform}(0, 1)$;

9      $L \leftarrow \{\min(l_1, l_2) | 1 \leq l_1, l_2 \leq \gamma, \epsilon_{l_1}^\tau > \frac{g_T(\hat{\tau}_{i+l_1}|\cdot)}{g_D(\hat{\tau}_{i+l_1}|\cdot)}, \epsilon_{l_2}^k > \frac{f_T(\hat{k}_{i+l_2}|\cdot)}{f_D(\hat{k}_{i+l_2}|\cdot)}\}$;

10      **if** $L < \gamma$ **then**

11          Sample $\hat{\tau}_{i+L} \sim g'(\tau_{i+L}|\cdot)$ and $\hat{t}_{i+L} = t_{i+L-1} + \hat{\tau}_{i+L}$;

12          Sample $\hat{k}_{i+L} \sim f'(k_{i+L}|\cdot)$;

13      $\mathcal{S} \leftarrow \mathcal{S} \cup \{(\hat{t}_{i+1}, \hat{k}_{i+1}), \ldots, (\hat{t}_{i+L}, \hat{k}_{i+L})\}$;

14      $i \leftarrow i + L$;

15      $t_i \leftarrow \hat{t}_i$;

16 $\mathcal{S} \leftarrow \{(t_i, k_i) \mid (t_i, k_i) \in \mathcal{S}, t_i \leq T\}$;

17 **Return:** $\mathcal{S}$

---

By default, we trained an 8-head, 20-layer target model and a 1-head, 1-layer draft model for each dataset. All models were trained using the Adam optimizer [11] for up to 1000 epochs with a batch size of 16 on one single NVIDIA RTX 4090. Early stopping based on validation log-likelihood was applied to prevent overfitting.

### C.3 Sampling Details

#### C.3.1 Sampling Synthetic TPPs

For the experiments in Section 5.2, we evaluate sampling performance within the time window $[0, 100]$. We report the discrepancy metrics $\Delta\mathcal{L}_{ar}^{syn}$ and $\Delta\mathcal{L}_{sd}^{syn}$, the KS statistics $D_{KS}$, the execution time $T_{AR}$ and $T_{SD}$, and the resulting speedup ratio $S_{AR/SD}$ as defined in Section 5.1. We conducted experiments across three random seeds and report the mean for each metric to ensure result robustness.

Apart from numerical metrics, we employ Kolmogorov-Smirnov plots (KS plots) to visualize sampling quality, which compare the empirical cumulative distribution $F_n(x)$ of the rescaled intervals $\{z_i\}_{i=1}^n$ against the theoretical cumulative distribution $F(x) = 1 - e^{-x}$ given by [2]. The KS plot visualizes points $\{(F(z_i), F_n(z_i))\}_{i=1}^n$, where perfect sampling would yield points along the 45-degree line.

We conduct formal statistical assessment using the Kolmogorov-Smirnov test with the following hypotheses:

$$H_0 : F_n(x) = F(x) \quad \text{versus} \quad H_1 : F_n(x) \neq F(x), \tag{25}$$

$$\text{Reject } H_0 \text{ if:} \quad D_{KS} > \frac{c(\alpha)}{\sqrt{n}}, \tag{26}$$

where $n$ is the number of sampled events, $D_{KS} = \sup_x |F_n(x) - F(x)|$ is the KS statistic, and $c(\alpha) = 1.36$ at significance level $\alpha = 0.05$ [13]. Based on this test, we construct a 95% confidence band $CB = \{(F(x), y) : y \in [F(x) - \frac{c(\alpha)}{\sqrt{n}}, F(x) + \frac{c(\alpha)}{\sqrt{n}}]\}$ for each KS plot. If the sampled events conform to the ground truth process at the 95% confidence level, all points in the KS plot should fall within this band.

We construct direct comparison visualizations between $F_n(x)$ and $F(x) = 1 - e^{-x}$ alongside KS plots across Transformer encoders (THP, SAHP, AttNHP) with draft length $\gamma = 10$ on three synthetic datasets. As evidenced in Fig. 4, $F_n(x)$ demonstrates strong concordance with $F(x) = 1 - e^{-x}$, and all points in the KS plots remain within the 95% confidence bands, providing robust statistical evidence for the distributional fidelity of our TPP-SD method.

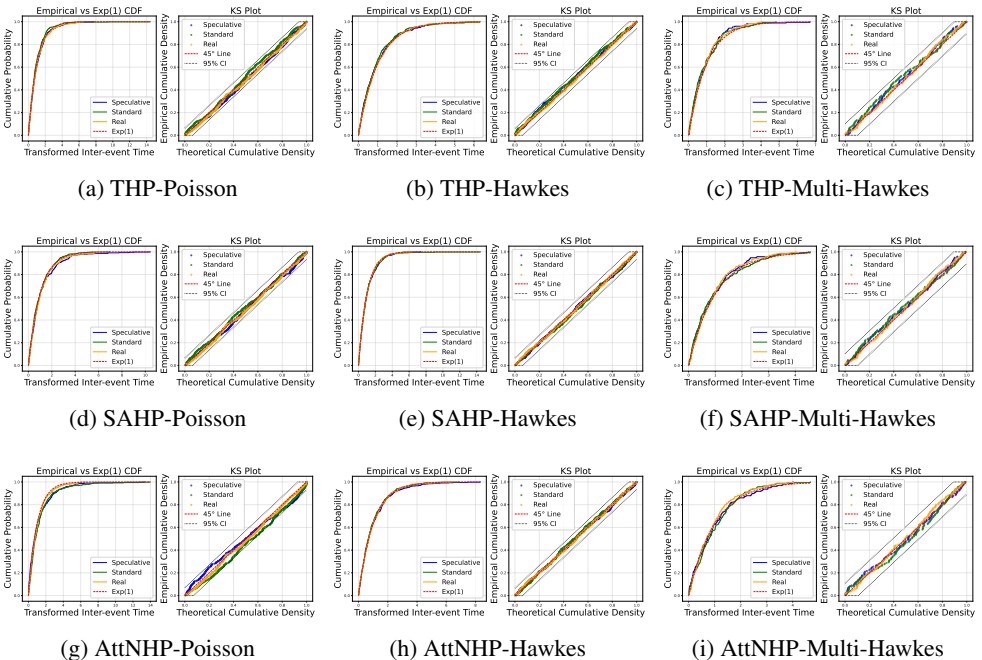

Figure 4: KS plots across THP (top row), SAHP (middle row), and AttNHP (bottom row) encoders with draft length $\gamma = 10$ on three synthetic datasets: Poisson (left column), Hawkes (middle column), and Multi-Hawkes (right column). Blue, green, and orange points represent samples from TPP-SD ($\gamma = 10$), AR sampling, and ground truth, respectively. Black dotted lines show 95% KS confidence bands.

### C.3.2 Sampling Real-world TPPs

For the experiments in Section 5.3, we conduct evaluations within the same time window of $[0, 100]$. In accordance with Appendix C.3.1, we report the discrepancy metric $\Delta\mathcal{L}^{\mathrm{real}}$, the Wasserstein distance $D_{\mathrm{WS}}^t$ and $D_{\mathrm{WS}}^k$, the execution time $T_{\mathrm{AR}}$ and $T_{\mathrm{SD}}$ for AR sampling and TPP-SD, and the resulting speedup ratio $S_{\mathrm{AR/SD}}$ defined in Section 5.1. For computation of $D_{WS}^t$ and $D_{WS}^k$, we set $M = 100, N = 100$, which means that we fix the first $M = 100$ events as history and perform $N = 100$ independent repetitions of sampling the 101-th event, obtaining $\{(t_i^{\mathrm{AR}}, k_i^{\mathrm{AR}}\}_{i=1}^{100}$ from AR sampling and $\{(t_i^{\mathrm{SD}}, k_i^{\mathrm{SD}}\}_{i=1}^{100}$ from TPP-SD, and subsequently computing $D_{\mathrm{WS}}^t$ and $D_{\mathrm{WS}}^k$ as elaborated in Section 5.1.

Beyond numerical metrics, we visualize event type distributions to qualitatively assess the sampling fidelity of TPP-SD on real datasets, where accurate event type sampling is critical. Figure 5 demonstrates that event type distributions from TPP-SD ($\{k_i^{\mathrm{SD}}\}$) consistently align with those from AR sampling ($\{k_i^{\mathrm{AR}}\}$) across all datasets and encoder architectures, confirming TPP-SD's ability to maintain distributional consistency with AR sampling in event type generation.

### C.4 Ablation Studies

We have investigated the impact of draft length $\gamma$ and draft model size on the sampling quality and sampling speed of TPP-SD with AttNHP encoder. Using the same sampling configuration and evaluation metrics as in Section 5.4, we also conducted experiments using THP and SAHP encoders.

**Draft Length.** As illustrated in Figs. 6a and 6b, experiments with THP and SAHP encoders reveal patterns consistent with our findings in Section 5.4: changes in draft length $\gamma$ minimally affect likelihood discrepancy $\Delta\mathcal{L}$ and distance metrics $D$, while acceptance rate $\alpha$ also diminishes with increasing $\gamma$, and speedup ratio $S_{\mathrm{AR/SD}}$ exhibits a characteristic peak followed by a decline, eventually dropping below $1\times$ when draft lengths become excessive. Intermediate draft lengths ($\gamma \approx 5$–$15$) optimize computational efficiency across all experimental configurations.

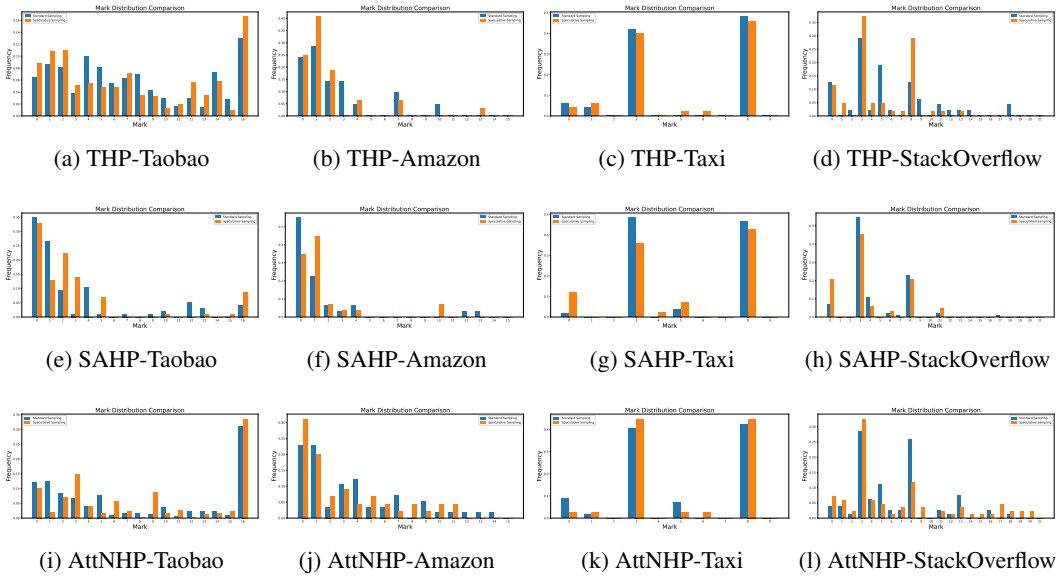

Figure 5: Event type histogram across THP (top row), SAHP (middle row), and AttNHP (bottom row) encoders with draft length $\gamma = 10$ on four real datasets: Taobao, Amazon, Taxi and StackOverflow. Blue bars and orange bars represent the frequency of sampled event types from AR sampling and TPP-SD.

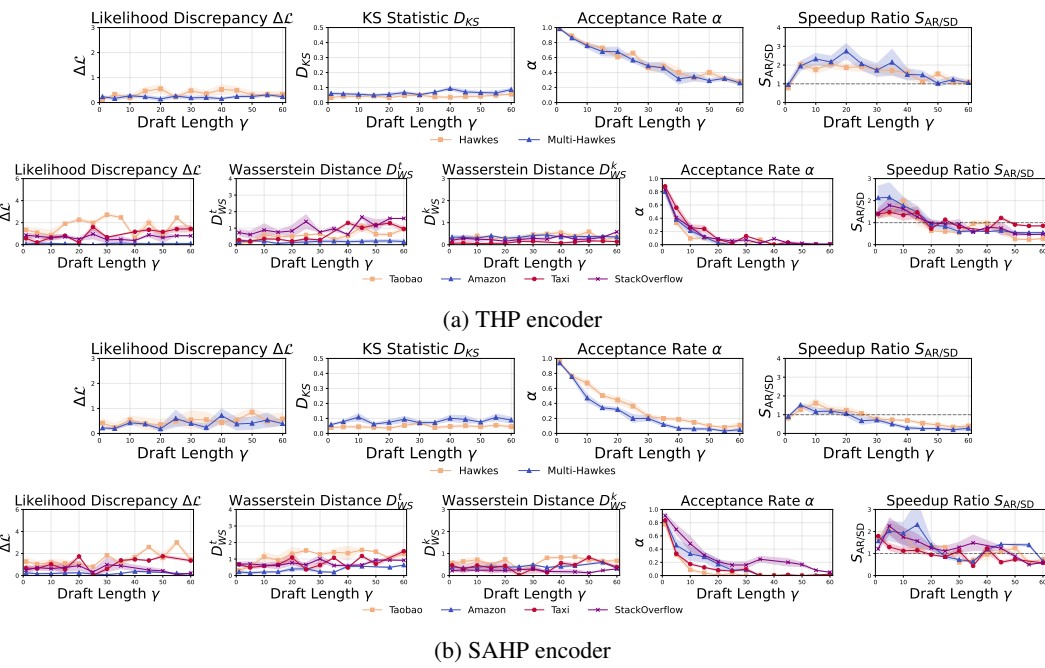

Figure 6: The impact of draft length on sampling quality measured by likelihood discrepancy ($\Delta\mathcal{L}$) and distance ($D_{KS}$ or $D_{WS}$), and on sampling speed measured up speedup ratio $S_{AR/SD}$. We conduct experiments across five random seeds with (a) THP encoder and (b) SAHP encoder, and report the average for each metric.

**Draft Model Size.** As evidenced by Table 4, increasing the draft model size with either THP or SAHP encoder preserves sampling quality (measured by $\Delta\mathcal{L}$ and $D$) and improves the acceptance rate $\alpha$, but reduces the speedup ratio $S_{\text{AR/SD}}$. The 1-head, 1-layer configuration remains optimal for the draft model across these encoder architectures.

Table 4: Performance of TPP-SD with draft length $\gamma = 10$ under different size of draft model. We use KS statistic ($D_{\text{KS}}$) for synthetic datasets and Wasserstein Distance ($D^t_{\text{WS}}, D^k_{\text{WS}}$) for real datasets. We conducted experiments across three random seeds and report the mean for each metric. For all metrics, the best result is shown in **bold**, and the second best result is shown in underline.

| Dataset | Encoder Type | Draft Model head | layer | $\Delta\mathcal{L}$ | $D_{\text{KS}}$ (↓) | $D^t_{\text{WS}}$ (↓) | $D^k_{\text{WS}}$ (↓) | $\alpha$ (↑) | $T_{\text{AR}}$ (↓) | $T_{\text{SD}}$ (↓) | $S_{\text{AR/SD}}$ (↑) |
|---|---|---|---|---|---|---|---|---|---|---|---|
| Multi-Hawkes | THP | 1 | 1 | 0.226 | 0.008 | - | - | 0.790 | 3.990 | **1.970** | **2.025** |
| | | 2 | 4 | 0.202 | 0.019 | - | - | 0.830 | 3.990 | 2.179 | 1.831 |
| | | 4 | 6 | **0.144** | **0.006** | - | - | **0.950** | 3.990 | 2.208 | 1.807 |
| Multi-Hawkes | SAHP | 1 | 1 | **0.054** | 0.008 | - | - | 0.640 | 3.543 | **2.143** | **1.653** |
| | | 2 | 4 | 0.143 | **0.004** | - | - | 0.740 | 3.543 | 2.669 | 1.327 |
| | | 4 | 6 | 0.395 | 0.008 | - | - | **0.800** | 3.543 | 2.734 | 1.296 |
| Taobao | THP | 1 | 1 | **0.033** | - | **0.175** | **0.222** | 0.18 | 5.890 | **3.460** | **1.702** |
| | | 2 | 4 | 0.307 | - | **0.175** | 0.339 | 0.210 | 5.890 | 3.900 | 1.510 |
| | | 4 | 6 | 0.409 | - | 0.291 | 0.301 | **0.240** | 5.890 | 4.920 | 1.197 |
| Taobao | SAHP | 1 | 1 | 0.410 | - | **0.302** | **0.630** | 0.14 | 2.460 | **1.621** | **1.518** |
| | | 2 | 4 | **0.334** | - | 0.403 | 0.742 | 0.19 | 2.460 | 1.850 | 1.330 |
| | | 4 | 6 | 0.486 | - | 0.545 | 0.745 | 0.21 | 2.460 | 1.990 | 1.236 |

# D Additional Discussions

## D.1 Why Not Use CIF-based Speculative Decoding?

In theory, CIF-based speculative decoding is feasible. For example, we can use a homogeneous Poisson process with constant intensity $\bar{\lambda}$ as the draft model to generate candidate timestamps $\{\tilde{t}_i\}$. These candidates are then input into a CIF-based Transformer point process to compute the CIF at each position, $\{\lambda^*(\tilde{t}_i)\}$, and the acceptance ratio $\{\lambda^*(\tilde{t}_i)/\bar{\lambda}\}$. A candidate timestamp $t_i$ is accepted if all previous timestamps are accepted and $\epsilon < \frac{\lambda^*(\tilde{t}_i)}{\bar{\lambda}}$, where $\epsilon \sim \text{Uniform}[0,1]$. However, it has two significant drawbacks: (1) Determining $\bar{\lambda}$ is challenging because we must ensure that $\bar{\lambda}$ is greater than the CIF at all candidate timestamps, and the CIF values are history-dependent and stochastic. To be safe, we generally have to use a relatively large $\bar{\lambda}$, which reduces the acceptance rate and leads to low sampling efficiency. (2) The CIF-based TPP-SD may fail to generate any valid timestamp in each iteration. If the first proposed timestamp is rejected, the CDF-based TPP-SD can sample a replacement from $g_T(\tau|\cdot)$, but the CIF-based method cannot do this. This can result in cases where a single forward pass of the target model fails to generate any valid timestamp, further reducing sampling efficiency.

## D.2 Necessity of Sampling from Large Target Models

A critical consideration in TPP sampling is the necessity of sampling from large draft models. For TPP data, large Transformer encoders (e.g., multi-layer, multi-head, with high-dimensional representations) are crucial for capturing complex temporal dependencies. Their larger parameter capacity allows them to model intricate data distributions and generate high-quality samples—albeit at the cost of slower autoregressive sampling. In contrast, smaller Transformer encoders (e.g., single-layer, single-head, with low-dimensional representations) are much faster at sampling due to reduced parameter capacity. However, they often struggle to model complex temporal patterns accurately, resulting in lower-quality samples that may not align well with the underlying data distribution.

To empirically demonstrate this performance gap, we evaluate a simple draft model $\mathcal{M}_D$ (single-layer, single-head THP) on both synthetic (Poisson, Hawkes, Multi-Hawkes) and real datasets (Taxi, StackOverflow, Amazon, Taobao), measuring sampling fidelity via $\Delta\mathcal{L}$ and $D_{KS}, D_{WS}$, and efficiency via wall-time.

As shown in Table 5, the simplistic architecture of $\mathcal{M}_D$ struggles to capture temporal dependencies effectively even on synthetic datasets. Performance degrades further on real datasets, with significantly

Table 5: Evaluation of sampling methods across synthetic and real-world datasets. Autoregressive sampling with the draft model (AR (draft)) consistently yields lower sample fidelity (higher $\Delta\mathcal{L}$, $D_{KS}$, and $D_{WS}$) compared to the target model (AR (target)) and our proposed TPP-SD. For each metric, the **best** result is bolded, and the second-best is underlined.

| Metric | Sampling Type | Poisson | Hawkes | Multi-Hawkes | Taxi | StackOverflow | Amazon | Taobao |
|---|---|---|---|---|---|---|---|---|
| $\Delta\mathcal{L}(\downarrow)$ | AR (draft) | 2.174 | 1.348 | 1.305 | 1.504 | 1.136 | 0.614 | 2.122 |
| | AR (target) | 0.542 | 0.753 | **0.022** | 0.441 | 0.587 | **0.056** | 0.446 |
| | TPP-SD | **0.349** | **0.276** | 0.321 | **0.065** | **0.231** | 0.129 | **0.033** |
| $D_{KS}(\downarrow)$ | AR (draft) | 0.153 | 0.141 | 0.126 | - | - | - | - |
| | AR (target) | 0.038 | 0.044 | 0.069 | - | - | - | - |
| | TPP-SD | **0.036** | **0.043** | **0.053** | - | - | - | - |
| $D^t_{WS}(\downarrow)$ | AR (draft) | - | - | - | 0.765 | 0.994 | 0.534 | 0.577 |
| | AR (target) | - | - | - | 0.201 | 0.470 | 0.189 | 0.236 |
| | TPP-SD | - | - | - | **0.082** | **0.391** | **0.078** | **0.076** |
| $D^k_{WS}(\downarrow)$ | AR (draft) | - | - | - | 0.768 | 0.821 | 0.759 | 1.384 |
| | AR (target) | - | - | - | **0.055** | 0.376 | **0.184** | **0.267** |
| | TPP-SD | - | - | - | 0.655 | **0.375** | 0.418 | 0.751 |
| Wall-time ($\downarrow$) | AR (draft) | **0.987** | **1.240** | **1.530** | **0.150** | **0.120** | **0.070** | **0.410** |
| | AR (target) | 3.477 | 5.147 | 4.007 | 1.157 | 1.353 | 1.023 | 5.890 |
| | TPP-SD | 1.647 | 2.547 | 1.893 | 0.453 | 0.700 | 0.290 | 3.460 |

larger $\Delta\mathcal{L}$ and $D_{WS}$ values, as $\mathcal{M}_D$ cannot model the complex dependencies present in the data. Draft model alone performs poorly on this task, demonstrating the necessity of a large target model and the importance of our proposed TPP-SD framework, which leverages the speed of the draft model without sacrificing the fidelity provided by the target model.

### D.3 Feasibility of Using Simple Poisson Draft

The standard thinning algorithm generates candidate events using a homogeneous Poisson process $\text{PoiP}(\bar{\lambda})$. This motivates a compelling question for our TPP-SD framework: is it possible to employ a draft model as simple as $\text{PoiP}(\bar{\lambda})$ to further accelerate sampling from complex target models without compromising fidelity?

We replace the 1-head-1-layer (abbreviated as 1-H-1-L) draft model to a simple homogeneous Poisson process draft model, whose parameter is estimated by maximum likelihood estimation on the data, i.e., $\hat{\lambda} = \frac{n}{T}$, where $n$ is the number of events and $T$ is the time span of the dataset. As for the experiment setup, we adopt THP as the backbone of target model for illustration simplicity. We conduct the additional experiments on 3 synthetic datasets (i.e., Poisson, Hawkes, and Multi-Hawkes) and 2 real-world datasets (i.e., Taobao and Amazon).

Since Poisson process is history-independent, we can sample the candidate inter-event times $\tau_i$ **all at once** from $\text{Exp}(\hat{\lambda})$. It is worth noting that for TPPs with marks (e.g. Multi-Hawkes), to ensure an efficient drafting process, we first estimate the mark distribution from the training data using MLE (i.e., $\hat{p}_m = \frac{n_m}{n}$, where $n_m$ is the number of events with mark $m$ and $n$ is the total number of events), and then sample the marks from the estimated mark distribution **all at once** as well.

We can see from Table 6 that, on synthetic datasets, TPP-SD with Poisson draft maintains high fidelity in sampling while achieving even more significant speedup across three datasets. The fast drafting process of Poisson compensates for the relatively lower acceptance rate. However, on real-world datasets, while TPP-SD with Poisson draft retains the sampling fidelity, we witness drop in speedup ratio. On complex real-world TPPs, the acceptance rate of Poisson draft is too low that it overwhelms the speedup from fast drafting. Therefore, in our setting of TPP-SD, we use a more robust drafting strategy, i.e., a slightly more complex draft model (1-layer-1-head THP) to ensure that TPP-SD can achieve high fidelity sampling across a wide range of datasets.

### D.4 Architectural Differences across Encoders

We incorporate the history encoders from THP, SAHP, and AttNHP into our CDF-based Transformer TPPs designed for TPP-SD. The architectural and implementation differences among these encoders may explain the varying sampling speed performance observed in Table 1 and Table 2.

Table 6: Comparison between a 1-head-1-layer (1-H-1-L) Transformer draft model and a simple Poisson process draft within the TPP-SD framework. On synthetic data, the Poisson draft achieves a higher speedup at the cost of fidelity. On real-world data, its low acceptance rate leads to inferior performance in both speed and fidelity. For each metric, the **best** result between the two TPP-SD methods is bolded, and the second-best is underlined.

| Dataset | Sampling Method | $\Delta\mathcal{L}(\downarrow)$ | $D_{KS}(\downarrow)$ | $D^t_{WS}(\downarrow)$ | $D^k_{WS}(\downarrow)$ | Acceptance Rate $\alpha(\uparrow)$ | Speedup $S_{AR/SD}(\uparrow)$ |
|---|---|---|---|---|---|---|---|
| Poisson | AR Sampling | 0.542 | 0.038 | - | - | - | - |
| | TPP-SD (1-H-1-L draft) | **0.349** | **0.036** | - | - | **0.740** | 2.110 |
| | TPP-SD (Poisson draft) | 0.630 | 0.050 | - | - | 0.370 | **3.230** |
| Hawkes | AR Sampling | 0.753 | 0.044 | - | - | - | - |
| | TPP-SD (1-H-1-L draft) | **0.276** | 0.043 | - | - | **0.720** | 2.113 |
| | TPP-SD (Poisson draft) | 0.286 | **0.036** | - | - | 0.520 | **5.430** |
| Multi-Hawkes | AR Sampling | 0.022 | 0.069 | - | - | - | - |
| | TPP-SD (1-H-1-L draft) | 0.321 | **0.053** | - | - | **0.750** | 2.117 |
| | TPP-SD (Poisson draft) | **0.268** | 0.057 | - | - | 0.390 | **3.610** |
| Taobao | AR Sampling | 0.446 | - | 0.236 | 0.267 | - | - |
| | TPP-SD (1-H-1-L draft) | **0.033** | - | **0.076** | 0.751 | **0.260** | **1.597** |
| | TPP-SD (Poisson draft) | 0.680 | - | 0.091 | **0.623** | 0.030 | 1.370 |
| Amazon | AR Sampling | 0.056 | - | 0.189 | 0.184 | - | - |
| | TPP-SD (1-H-1-L draft) | 0.129 | - | **0.078** | 0.418 | **0.270** | **3.550** |
| | TPP-SD (Poisson draft) | **0.066** | - | 0.090 | **0.086** | 0.010 | 2.230 |

Following the notation in Section 4.2, the temporal encoding $\mathbf{z}(t_i) \in \mathbb{R}^D$ for each encoder differs slightly:

$$\text{THP: } [\mathbf{z}(t_i)]_j = \begin{cases} \sin(t_i/10000^{j/D}) & j \text{ is even} \\ \cos(t_i/10000^{(j-1)/D}) & j \text{ is odd} \end{cases}, \tag{27}$$

$$\text{SAHP: } [\mathbf{z}(t_i)]_j = \begin{cases} \sin(j/10000^{j/D} + w_j t_i) & j \text{ is even} \\ \cos(j/10000^{(j-1)/D} + w_j t_i) & j \text{ is odd} \end{cases}, \tag{28}$$

$$\text{AttNHP: } [\mathbf{z}(t_i)]_j = \begin{cases} \sin(t_i/m \cdot (5M/m)^{j/D}) & j \text{ is even} \\ \sin(t_i/m \cdot (5M/m)^{(j-1)/D}) & j \text{ is odd} \end{cases}, \tag{29}$$

where $[\mathbf{z}(t_i)]_j$ is the $j$-th dimension of the temporal embedding vector $\mathbf{z}(t_i)$, and $M, m$ are hyperparameters in AttNHP.

The attention mechanisms differ across encoder architectures. For clarity, we present the single-head attention formulation for each model. The layer-$l$ history embedding of event $(t_i, k_i)$ is computed as:

$$\text{THP/SAHP: } \mathbf{h}^{(l)}(t_i) = \mathbf{h}^{(l-1)}(t_i) + \sum_{j=1}^{i} \frac{f(\mathbf{q}^{(l)}(t_j), \mathbf{k}^{(l)}(t_i))\mathbf{v}^{(l)}(t_j)}{\sum_{j=1}^{i} f(\mathbf{q}^{(l)}(t_j), \mathbf{k}^{(l)}(t_i))}, \tag{30}$$

$$\text{AttNHP: } \mathbf{h}^{(l)}(t_i) = \mathbf{h}^{(l-1)}(t_i) + \tanh\left(\sum_{j=1}^{i} \frac{f(\mathbf{q}^{(l)}(t_j), \mathbf{k}^{(l)}(t_i))\mathbf{v}^{(l)}(t_j)}{1 + \sum_{j=1}^{i} f(\mathbf{q}^{(l)}(t_j), \mathbf{k}^{(l)}(t_i))}\right), \tag{31}$$

where $f(\cdot, \cdot)$ is a Gaussian kernel $f(\mathbf{q}^{(l)}(t_j), \mathbf{k}^{(l)}(t_i)) = \exp(\frac{1}{\sqrt{D}}\mathbf{q}^{(l)}(t_i)^\top \mathbf{k}^{(l)}(t_j)) \in \mathbb{R}$. The key distinction lies in how query, key, and value vectors are derived at layer-$l$. In THP and SAHP, $\mathbf{q}^{(l)}(t_i) = \mathbf{Q}^{(l)}\mathbf{h}^{(l-1)}, \mathbf{k}^{(l)}(t_i) = \mathbf{K}^{(l)}\mathbf{h}^{(l-1)}, \mathbf{v}^{(l)}(t_i) = \mathbf{V}^{(l)}\mathbf{h}^{(l-1)}$ are obtained through linear projection, whereas in AttNHP:

$$\mathbf{q}^{(l-1)}(t_i) = \mathbf{Q}^{(l)}\text{concat}(1; \mathbf{z}(t_i)^\top; \mathbf{h}^{(l-1)}(t_i)^\top)^\top, \tag{32}$$

$$\mathbf{k}^{(l-1)}(t_i) = \mathbf{K}^{(l)}\text{concat}(1; \mathbf{z}(t_i)^\top; \mathbf{h}^{(l-1)}(t_i)^\top)^\top, \tag{33}$$

$$\mathbf{v}^{(l-1)}(t_i) = \mathbf{V}^{(l)}\text{concat}(1; \mathbf{z}(t_i)^\top; \mathbf{h}^{(l-1)}(t_i)^\top)^\top, \tag{34}$$

where $\mathbf{Q}^{(l)}, \mathbf{K}^{(l)}, \mathbf{V}^{(l)} \in \mathbb{R}^{D \times (2D+1)}$. AttNHP's complex attention mechanism doubles the dimension of intermediate vectors, particularly in multi-head configurations where each head requires

separate query, key, and value transformations. This architectural complexity results in significantly higher AR sampling latency compared to other models, making AttNHP an ideal candidate for acceleration via TPP-SD, where it achieves the largest speedup. Despite architectural similarities between THP and SAHP, implementation differences result in SAHP consistently achieving the shortest wall-time for both AR sampling and TPP-SD, leaving less room for additional acceleration.

### D.5 Differences between TPP Sampling and Long-horizon Prediction

While both TPP sampling and long-horizon prediction [35, 24] involve generating sequences of events over a continuous time interval given some history, they have fundamental differences. TPP sampling focuses on generating events from the correct distribution. Since sampling is inherently stochastic, comparing the exact events sampled with specific test set events isn't meaningful - the same distribution that we sample from can produce different event in each iteration. What matters is whether the sampled sequences follow the correct statistical patterns. On synthetic datasets, we evaluate the distributional conformity of sampled sequences from TPP-SD to the ground truth using [2] and KS statistic. On real datasets, we measure the distributional discrepancy between the sampled sequences from autoregressive sampling and TPP-SD with Wasserstein distance.

In contrast, long-horizon prediction specifically aims to forecast future events that closely match the ground truth on the test set. It's evaluated by metrics such as RMSE and optimal transport distance (OTD) that directly measure the discrepancy between predicted sequences and test-set sequences.

### D.6 Limitations

Compared to the thinning algorithm, our proposed TPP-SD is deep-learning based which requires large amount of event sequence data to learn the model's parameters. Therefore, the model is less suitable for sampling in data-scarce scenarios.

Besides, TPP-SD adopts the original SD framework from the LLM domain and requires deploying two models at the same time for accelerating sampling, which is slightly more computational demanding and less convenient. Recent advances in SD techniques [3, 16] can be incorporated to further optimize TPP-SD, and we leave this as future work.

