# OpenReview forum: "TPP-SD: Accelerating Transformer Point Process Sampling with Speculative Decoding"
_NeurIPS.cc/2025/Conference — NeurIPS 2025 poster_

### Official Review · Reviewer_yB4e · 2025-06-11

**Clarity:** 3
**Significance:** 3
**Originality:** 3
**Rating:** 5
**Confidence:** 4

**Summary:**

The paper introduces TPP-SD, a speculative decoding framework for accelerating Transformer-based temporal point process (TPP) sampling by drawing inspiration from the structural parallels between thinning algorithms and speculative decoding in language models. By employing a lightweight draft model to generate multiple candidate events and leveraging a larger target model for joint verification of event intervals and types through a mixture of log-normal CDFs and categorical distributions, TPP-SD achieves significant speedup—2–6× across synthetic and real datasets—while maintaining distributional fidelity via adaptive rejection sampling and an adjusted distribution mechanism.

**Questions:**

1.After drafting candidate events autoregressively, how is the target model ensured to verify all candidates in parallel? In the speculative decoding of LLM, a common practice is to concatenate all candidates into a single vector and enable parallel verification of the target model by modifying the attention map. Does TPP-SD adopt a similar approach? I would appreciate it if you could provide a more detailed explanation.


2.In Table 1 and Table 2, for metrics such as Likelihood Discrepancy, KS statistic, and Wasserstein Distance, TPP-SD sometimes achieves better values than AR Sampling. Why does TPP-SD sometimes exhibit higher sample quality than AR Sampling? In my understanding, TPP-SD employs a draft model with fewer parameters, which achieves acceleration while minimizing the sacrifice in output accuracy. However, the current observation that TPP-SD sometimes even attains higher output accuracy has left me confused. If there are any underlying issues or nuances I may have overlooked, I would appreciate it if you could clarify or point them out.

**Ethical Concerns:**

["NO or VERY MINOR ethics concerns only"]

**Final Justification:**

The authors have addressed my concerns in the rebuttal
Strength:

1.First to integrate Speculative Decoding with TTP sampling, offering a new research direction.

2.Links TPP thinning to speculative decoding, enabling an adaptive verifier that preserves the exact joint discrete/continuous output distribution.

3.Experiments on synthetic and real data confirm speed-ups with negligible loss in accuracy.

**Limitations:**

The authors have listed the limitations of the current work in Section D.4 Limitations.

**Paper Formatting Concerns:**

No major issues found in this paper.

**Quality:**

4

**Strengths And Weaknesses:**

Strengths:

1.This work applies Speculative Decoding to TTP sampling for the first time, which has the potential to inspire subsequent research.

2.This work cleverly uncovers the structural similarities between thinning algorithms for TPPs and speculative decoding for language models. It further demonstrates the adaptive verification and resampling strategy that maintains the same output distribution as autoregressive sampling when both discrete and continuous distributions coexist.

3.Sufficient experiments on both synthetic and real datasets demonstrate the acceleration effect, while also verifying that the impact on sampling accuracy is minimal.

Weaknesses:

1.In Section 4.2, formulas are listed to describe the structure of CDF-based Transformer TPPs, which is somewhat complex. Inserting a diagram here would enhance readability and clarify the structural details, making it easier for readers to follow.

2.The experimental section does not compare the sample quality and sampling speed of TPP-SD with other related TPP works under similar settings, which fails to intuitively demonstrate the advantages of TPP-SD compared to existing approaches.

---

> ### Author Rebuttal · Authors · 2025-07-29
>
> We sincerely thank you for your precious time and valuable comments. We are encouraged by your positive comments on our research novelty, technical soundness and extensive experiments, especially the recognition of unveiling the structural similarity between thinning algorithm and speculative decoding. We hope that our following clarifications can address your concerns.
>
> > **Q1: In Section 4.2, formulas are listed to describe the structure of CDF-based Transformer TPPs, which is somewhat complex. Inserting a diagram here would enhance readability and clarify the structural details, making it easier for readers to follow.**
>
> **A1:** Sure, we will include a diagram in Section 4.2 to visually represent the structure of CDF-based Transformer TPPs in the camera-ready version.
>
> > **Q2: The experimental section does not compare the sample quality and sampling speed of TPP-SD with other related TPP works under similar settings, which fails to intuitively demonstrate the advantages of TPP-SD compared to existing approaches.**
>
> **A2:** To the best of our knowledge, existing literature in the community of TPP has primarily focused on improving model fitting and parameter estimation, with little emphasis on sampling acceleration. TPP-SD is the first work to accelerate high-performance Transformer-based TPP sampling. As such, there are no existing works that provide a direct comparison with TPP-SD.
>
> > **Q3: After drafting candidate events autoregressively, how is the target model ensured to verify all candidates in parallel? In the speculative decoding of LLM, a common practice is to concatenate all candidates into a single vector and enable parallel verification of the target model by modifying the attention map. Does TPP-SD adopt a similar approach? I would appreciate it if you could provide a more detailed explanation.**
>
> **A3:** Similar to [1], we adopt a similar parallel verification approach. We concatenate candidate events with the prefix (generated events) into a single vector and feed it to the target model. During parallel verification of the target model, we maintain the causal attention map (lower triangular matrix) to ensure that when computing probabilities for candidate events, each event strictly attends to previous events in the draft sequence.
>
> > **Q4: In Table 1 and Table 2, for metrics such as Likelihood Discrepancy, KS statistic, and Wasserstein Distance, TPP-SD sometimes achieves better values than AR Sampling. Why does TPP-SD sometimes exhibit higher sample quality than AR Sampling? In my understanding, TPP-SD employs a draft model with fewer parameters, which achieves acceleration while minimizing the sacrifice in output accuracy. However, the current observation that TPP-SD sometimes even attains higher output accuracy has left me confused. If there are any underlying issues or nuances I may have overlooked, I would appreciate it if you could clarify or point them out.**
>
> **A4:** This is a keen observation. The reviewer is correct that the slight variations in sample quality metrics are indeed due to the stochasticity of the sampling process over a finite number of runs. Though it seems that TPP-SD sometimes achieves better scores than AR sampling (or vice versa) on $\Delta \mathcal{L}$, $D_{KS}$, or $D_{WS}$, these differences does not imply that one method is superior to the other in sampling fidelity. As long as these metrics for both methods are statistically close to zero, it empirically demonstrates that sampled events from TPP-SD are distributionally identical to those from AR sampling, i.e., preserving the target model's high fidelity.
>
> We also want to emphasize that our TPP-SD method does not involve any "sacrifice" in generative quality. The verification step is theoretically guaranteed to ensure that the distribution of the final output samples matches that of standard autoregressive sampling.
>
> References:
>
> [1] Fast Inference from Transformers via Speculative Decoding, ICML, 2023.

---

### Official Review · Reviewer_k4Sa · 2025-07-02

**Clarity:** 4
**Significance:** 3
**Originality:** 3
**Rating:** 5
**Confidence:** 4

**Summary:**

The paper proposes TPP-SD by connecting thinning in Temporal Point Process and Speculative Decoding. By unifying the two processes in the same proposal-verification framework, the paper introduces a draft model for the proposal distribution in the thinning process. The paper also extends the original speculative decoding algorithm to rejection sampling on continuous CDFs. The experiments support the authors' claim that TPP-SD can achieve sampling speedup with small degradation to quality on both synthetic TPP data and real time series data.

**Questions:**

- One baseline I'm curious about is a thinning baseline: What happens if the proposal distribution is something simple, such as a Poisson process? This is a single-parameter draft model, where you can estimate the parameter with MLE on the data. When evaluation computation is not the bottleneck, this method should also provide a lot of speedup.

**Ethical Concerns:**

["NO or VERY MINOR ethics concerns only"]

**Final Justification:**

The authors have addressed my concerns in the rebuttal and I felt confident to accept this paper.

**Limitations:**

yes

**Quality:**

4

**Strengths And Weaknesses:**

# Quality
- The paper is technically sound, and the claims are well supported with theory and experiments.

# Clarity
- The paper is very clearly written. I especially enjoyed the section that draws the connection between TPP and SD and found it highly educational.

# Significance
Strengths:
- While the techniques (thinning and Speculative Decoding) are not new, the paper points out the connection between thinning and SD, which is novel to the audience who might be familiar with individual subfields. This is very significant since it will encourage more cross-pollination between the fields of TPP and LLM sampling. This alone makes me feel inclined to accept this paper.
- The experiments support the authors' claims very well, and the theory behind the algorithm is sound.

Weaknesses:
- The experiments lack some important baseline for it to be practical, namely, directly using a simple TPP as the draft model. See the questions section for more details. This causes me to hesitate from giving a strong acceptance, but I'm happy to increase my score if additional experiments are conducted.

# Originality
- The connection between TPP and SD is very novel, and this is the first time SD is applied to TPP as far as I know (low confidence).

---

> ### Author Rebuttal · Authors · 2025-07-29
>
> We are sincerely grateful for your enthusiastic and positive feedback on our technical soundness, insightful investigations, extensive experiments, and good writing, especially for recognizing the novelty of TPP-SD in connecting thinning algorithm with speculative decoding. We hope that our following clarifications and new experiments can address your concerns.
>
> > **Q1: What happens if the proposal distribution is something simple, such as a Poisson process? This is a single-parameter draft model, where you can estimate the parameter with MLE on the data. When evaluation computation is not the bottleneck, this method should also provide a lot of speedup.**
>
> **A1:** This is a very interesting suggestion.
>
> We replace the 1-head-1-layer (abbreviated as 1-H-1-L) draft model to a simple **homogeneous Poisson process** draft model, whose parameter is estimated by MLE on the data, i.e., $\hat\lambda = \frac{n}{T}$, where $n$ is the number of events and $T$ is the time span of the dataset.
> As for the experiment setup, we adopt THP as the backbone of target model for illustration simplicity. We conduct the additional experiments on 3 synthetic datasets (i.e., Poisson, Hawkes, and Multi-Hawkes) and 2 real-world datasets (i.e., Taobao and Amazon).
>
> Since Poisson process is history-independent, we can sample the candidate inter-event times $\tau_i$ **all at once** from $\text{Exp}(\hat\lambda)$. It is worth noting that for TPPs with marks (e.g. Multi-Hawkes), to ensure an efficient drafting process, we first estimate the mark distribution from the training data using MLE (i.e., $\hat{p}_m = \frac{n_m}{n}$, where $n_m$ is the number of events with mark $m$ and $n$ is the total number of events), and then sample the marks from the estimated mark distribution **all at once** as well.
>
> We can see from the table that: on synthetic datasets, TPP-SD with Poisson draft maintains high fidelity in sampling while achieving even more significant speedup across three datasets. The fast drafting process of Poisson compensates for the relatively lower acceptance rate. However, on real-world datasets, while TPP-SD with Poisson draft retains the sampling fidelity, we witness drop in speedup ratio. On complex real-world TPPs, the acceptance rate of Poisson draft is too low that it overwhelms the speedup from fast drafting. Therefore, in our original setting of TPP-SD, we use a more robust drafting strategy, i.e., a slightly more complex draft model (1-layer-1-head THP) to ensure that TPP-SD can achieve high fidelity sampling across a wide range of datasets.
>
> This question is highly insightful and has greatly inspired us. We will include this additional experiment in the camera-ready version.
>
> | Dataset      | Sampling Method               | $\Delta \mathcal{L}$ (↓) | $D_{KS}$ (↓) | $D_{WS}^t$ (↓) | $D_{WS}^k$ (↓) | Acceptance Rate $\alpha$ (↑) | Speedup Ratio $S_{AR/SD}$ (↑) |
> | ------------ | ----------------------------- | ------------------------ | ------------ | -------------- | -------------- | ---------------------------- | ----------------------------- |
> | Poisson      | AR Sampling                   | 0.542                    | 0.038        | -              | -              | -                            | -                             |
> |              | TPP-SD (1-H-1-L draft) | 0.349                    | 0.036        | -              | -              | 0.740                        | 2.110                         |
> |              | TPP-SD (Poisson draft)        | 0.630                    | 0.050        | -              | -              | 0.370                        | **3.230**                     |
> | Hawkes       | AR Sampling                   | 0.753                    | 0.044        | -              | -              | -                            | -                             |
> |              | TPP-SD (1-H-1-L draft) | 0.276                    | 0.043        | -              | -              | 0.720                        | 2.113                         |
> |              | TPP-SD (Poisson draft)        | 0.286                    | 0.036        | -              | -              | 0.520                        | **5.430**                     |
> | Multi-Hawkes | AR Sampling                   | 0.022                    | 0.069        | -              | -              | -                            | -                             |
> |              | TPP-SD (1-H-1-L draft) | 0.321                    | 0.053        | -              | -              | 0.750                        | 2.117                         |
> |              | TPP-SD (Poisson draft)        | 0.268                    | 0.057        | -              | -              | 0.390                        | **3.610**                     |
> | Taobao       | AR Sampling                   | 0.446                    | -            | 0.236          | 0.267          | -                            | -                             |
> |              | TPP-SD (1-H-1-L draft) | 0.033                    | -            | 0.076          | 0.751          | 0.260                        | **1.597**                     |
> |              | TPP-SD (Poisson draft)        | 0.680                    | -            | 0.091          | 0.623          | 0.030                        | 1.370                         |
> | Amazon       | AR Sampling                   | 0.056                    | -            | 0.189          | 0.184          | -                            | -                             |
> |              | TPP-SD (1-H-1-L draft) | 0.129                    | -            | 0.078          | 0.418          | 0.270                        | **3.550**                     |
> |              | TPP-SD (Poisson draft)        | 0.066                    | -            | 0.090          | 0.086          | 0.010                        | 2.230                         |

---

### Official Review · Reviewer_RM4F · 2025-07-02

**Clarity:** 2
**Significance:** 3
**Originality:** 2
**Rating:** 4
**Confidence:** 4

**Summary:**

This paper introduces TPP-SD, an approach that accelerates sampling in Transformer-based temporal point processes by adapting speculative decoding techniques originally developed for large language models. The core idea is to use a smaller draft model to generate multiple candidate events, which are then validated in parallel by a larger, more accurate target model, leveraging the similarities between the thinning algorithm for point processes and speculative decoding for autoregressive generative models. The authors design a CDF-based Transformer TPP amenable to the speculative decoding mechanism, propose an efficient algorithm for continuous-time event sampling, and evaluate the approach on both synthetic and real-world datasets. The experimental results indicate that TPP-SD achieves 2–6× speedups compared to standard autoregressive sampling, while producing event sequences virtually indistinguishable in distribution from those produced by the standard method. The authors further investigate the impact of draft length and draft model size on quality and efficiency.

**Questions:**

Q1: In theory, the output distributions of TTP-SD and AR sampling should be identical, as the author acknowledges in Section 5.3. So why do these two methods result in differences in fidelity? If these differences arise from stochastic fluctuations, then reporting these metrics might lack the necessary motivation.

Q2: Does TTP-SD and AR sampling support parallel inference in batch samples, and in this case, will the speedup be limited?

**Ethical Concerns:**

["NO or VERY MINOR ethics concerns only"]

**Final Justification:**

Strengths
- The paper provides a well-grounded adaptation of speculative decoding to TPP sampling. The mathematical treatment of the adjusted distribution for continuous variables, as well as the connection to LLM speculative decoding, are clearly motivated, and theoretical correctness proofs are outlined in the main text (Section 4.3) and referenced in the Appendix.

Weaknesses
- While the analogy between thinning and speculative decoding is insightful, the core algorithmic advance is a relatively direct adaptation of existing SD techniques from the LLM domain.

- The author lacks an explanation of the acceleration-related background in related work (in the AR sampling or TTP fields). This also affects the choice of baseline, as there is a lack of comparison with acceleration techniques related to AR models, such as RNN/LSTM, which can provide acceleration but pose a risk of performance loss.

**Limitations:**

yes

**Quality:**

3

**Strengths And Weaknesses:**

Summary Of Strengths
- Technical Soundness and Theoretical Clarity: The paper provides a well-grounded adaptation of speculative decoding to TPP sampling. The mathematical treatment of the adjusted distribution for continuous variables, as well as the connection to LLM speculative decoding, are clearly motivated, and theoretical correctness proofs are outlined in the main text (Section 4.3) and referenced in the Appendix.
- Clarity and Organization: The paper is mostly well-structured and the technical writing is generally clear, with necessary background provided in Section 2 and related work in Section 3. The pseudo-code and algorithm descriptions, while referenced as being in the appendix, are described concisely in the main paper.

Summary Of Weaknesses
- Limited Novelty Beyond Application: While the analogy between thinning and speculative decoding is insightful, the core algorithmic advance is a relatively direct adaptation of existing SD techniques from the LLM domain. The paper does not extensively discuss unique challenges or TPP-specific adaptations (beyond continuous CDF normalization) that set it apart as a fundamentally new algorithm.
- Limited Baseline: The author lacks an explanation of the acceleration-related background in related work (in the AR sampling or TTP fields). This also affects the choice of baseline, as there is a lack of comparison with acceleration techniques related to AR models, such as RNN/LSTM, which can provide acceleration but pose a risk of performance loss.

---

> ### Author Rebuttal · Authors · 2025-07-29
>
> We are genuinely grateful for your valuable feedback. We are encouraged by your positive comments on our technical soundness, overall clarity and organization. We sincerely hope our clarifications below can address your concerns.
>
> > **Q1: Limited Novelty Beyond Application: While the analogy between thinning and speculative decoding is insightful, the core algorithmic advance is a relatively direct adaptation of existing SD techniques from the LLM domain. The paper does not extensively discuss unique challenges or TPP-specific adaptations (beyond continuous CDF normalization) that set it apart as a fundamentally new algorithm.**
>
> **A1:** While we agree that our work is inspired by the framework of SD from the LLM domain, we would like to clarify that this process is not a straightforward adaptation, and requires overcoming fundamental challenges unique to TPPs.
>
> - Unlike LLMs, which only involve sampling from discrete token spaces, TPPs require sampling both continuous timestamps and discrete event types. While the sampling of discrete event types is analogous to token generation in LLMs, the continuous timestamp sampling is a key distinction and a unique challenge specific to TPPs.
>
> In particular, our method must sample from adjusted continuous distributions after rejection (see Equation 3), which is not addressed in prior SD literature. To handle this, we propose a novel acceptance-rejection mechanism (Theorem 1) that enables efficient sampling without requiring expensive numerical integration. This constitutes a fundamental advancement in applying speculative decoding to continuous variable generation and is essential for enabling high-fidelity, efficient sampling in TPPs.
>
> - Existing Transformer TPPs[2,3,4] use conditional intensity function (CIF) parameterization, fundamentally incompatible with SD (Appendix D.1). CDF-based alternatives are easier to sample from, but the mainstream implementation[1] relies on the RNN architecture, which is unsuitable for parallel verification. We therefore designed a CDF-based Transformer architecture (Section 4.2) enabling both high-quality modeling and SD compatibility—a non-trivial contribution requiring careful design.
>
> > **Q2: Limited Baseline: The author lacks an explanation of the acceleration-related background in related work (in the AR sampling or TPP fields). This also affects the choice of baseline, as there is a lack of comparison with acceleration techniques related to AR models, such as RNN/LSTM, which can provide acceleration but pose a risk of performance loss.**
>
> **A2:** We appreciate the reviewer's comment and **will include acceleration-related work on AR sampling** (such as advances in SD in LLM domain). To the best of our knowledge, our work is the first systematic approach to accelerating Transformer-based TPP sampling while preserving distributional fidelity, as the TPP literature has primarily focused on model expressiveness and fitting quality rather than sampling efficiency.
>
> While using a recurrent architecture like LSTM represents one possible approach to the speed-performance trade-off, our work is motivated by a different premise. The community has validated Transformer TPPs' superior performance in capturing complex dependencies [2,3,4]. Rather than re-evaluating architectural trade-offs, we aimed to enable significant acceleration without compromising the high generative fidelity that makes these SOTA models preferable in the first place.
>
> > **Q3: In theory, the output distributions of TPP-SD and AR sampling should be identical, as the author acknowledges in Section 5.3. So why do these two methods result in differences in fidelity? If these differences arise from stochastic fluctuations, then reporting these metrics might lack the necessary motivation.**
>
> **A3:** The minor differences in fidelity metrics are indeed due to stochastic fluctuations inherent in the sampling process. Our motivation for reporting $\Delta \mathcal{L}$ and $D_{KS}$ / $D_{WS}$ is to provide a empirical validation that samples generated by TPP-SD are identical in distribution to those generated by AR sampling. As long as these metrics are statistically close to zero, we demonstrate that TPP-SD achieves speedup without distorting the sampling distribution.
>
> > **Q4: Does TPP-SD and AR sampling support parallel inference in batch samples, and in this case, will the speedup be limited?**
>
> **A4:** We have not explored batch inference for TPP-SD in this work, as we follow the standard SD framework introduced in [5], which operates with a batch size of 1.
>
> In the speculative decoding literature, enabling batch inference (i.e., batch size \( > 1 \) remains an active research topic due to the inherent challenge of synchronizing multiple sequences within a batch. This complexity also applies to TPP-SD, where each sequence may have different lengths and rejection patterns, making parallel decoding non-trivial.
>
> We consider extending TPP-SD to support efficient batch inference a promising direction for future work.
>
> References:
>
> [1] Intensity-Free Learning of Temporal Point Processes, ICLR, 2020.
>
> [2] Transformer Hawkes Process, ICML, 2020.
>
> [3] Self-Attentive Hawkes Process, ICML, 2020.
>
> [4] Transformer Embeddings of Irregularly Spaced Events and Their Participants, ICLR, 2022.
>
> [5] Fast Inference from Transformers via Speculative Decoding, ICML, 2023.

---

> > ### Comment · Reviewer_RM4F · 2025-08-05
> >
> > Thanks the authors for the response. I agree that there are some issue to solve for sampling in the continuous space. My concern is how important it is and how it benefits the other research in the field.
> >
> > I think the paper would be further strengthened if the above point could be discussed more in the paper and the required experiments are included in the paper as well.
> >
> > As for the score, I tend to keep my original decision.

---

> ### Author Response · Authors · 2025-08-06
> **Clarification on the Significance and Broader Impact of Our Work**
>
> Dear Reviewer RM4F,
>
> Thank you for your insightful feedback. We appreciate you highlighting the need to better articulate the importance of our work and its benefits to the broader research field, and we agree that this is crucial for strengthening the paper.
>
> We would like to clarify the significance of accelerating TPP sampling. The primary impact lies in making **large-scale simulation and data synthesis** practical for complex Transformer-based TPPs, where sampling efficiency is a major bottleneck.
>
> In research fields like computational neuroscience, simulating neural spike trains is fundamental for **validating hypotheses and understanding neural dynamics**. Similarly, in seismology or epidemiology, **simulating future event cascades** requires generating a vast number of sequences. The slow autoregressive sampling of powerful Transformer TPPs makes such large-scale studies computationally prohibitive. TPP-SD lowers the barrier by offering a maximum of 6x speedup, enabling researchers to conduct larger-scale simulations for downstream tasks with significantly shorter time.
>
> In real-world scenarios such as finance and healthcare, event sequence data (e.g., high-frequency transactions or patient hospital arrivals) can be scarce, imbalanced, or privacy-sensitive. Large-scale sampling from a well-trained TPP model allows for the creation of high-fidelity synthetic datasets that share the same statistical properties as the original data. These synthetic samples can be used for **downstream tasks like model training, system stress-testing, or public data release** without compromising privacy. The efficiency of TPP-SD makes this data synthesis process practical, especially when generating long sequences with high event frequency.
>
> We also wish to state the generalizability of our proposed continuous-space sampling mechanism. It is an important mid-way module to enable speculative decoding for any continuous-time Transformer, which we believe is overlooked in prior SD or continuous-time transformer acceleration literature.
>
> We are grateful for your suggestion. We will further elaborate on why efficient sampling is a critical yet overlooked problem in the TPP community and provide case studies on how TPP-SD benefits other research in the camera-ready version. Thank you again for helping us improve the paper.
>
> Best regards,
>
> Authors of Submission 6980

---

### Official Review · Reviewer_HqH3 · 2025-07-03

**Clarity:** 3
**Significance:** 3
**Originality:** 2
**Rating:** 3
**Confidence:** 2

**Summary:**

This paper proposes TPP-SD, a novel method that accelerates Transformer-based temporal point-process sampling through speculative decoding. The framework first leverages a smaller draft model to generate multiple candidate events, then uses a larger target model to verify them. Experiments demonstrate a 2 – 6 × speed-up on both synthetic and real datasets.

**Questions:**

- Can you report the performance of small model, such as the draft model on those datasets?

**Ethical Concerns:**

["NO or VERY MINOR ethics concerns only"]

**Limitations:**

yes

**Quality:**

2

**Strengths And Weaknesses:**

**Strength**

- The proposed approach achieves 2 – 6× faster sampling with virtually no loss in ΔL or Wasserstein quality.

- The authors conduct extensive experiments span both synthetic and real datasets, include ablations over γ, draft size, and event-type count K, and consistently confirm the speed–quality trade-off.

**Weaknesses**

- Limited baseline: The authors do not report the performance/speed of draft model on those datasets. It is not clear whether the draft model can already work well on this task efficiently. Notably, as show in Table 3, it seems speculative decoding with smaller draft may achieve better speed up. Therefore, it seems smaller model such as draft model size can also work well on this task.

- The proposed approach does not appear practical for real-world applications. Table 2 shows that the speed-up diminishes as K increases, and in real-world settings K is typically much larger. It is therefore unclear whether the method would remain effective in such scenarios.

---

> ### Author Rebuttal · Authors · 2025-07-29
>
> We sincerely thank you for your valuable comments and insightful questions. We are encouraged by your recognition on the fidelity-preserving ability and the efficiency of TPP-SD. We hope that our following clarifications and new experiments can address your concerns.
>
> > **Q1: Limited baseline: The authors do not report the performance/speed of draft model on those datasets. It is not clear whether the draft model can already work well on this task efficiently. Notably, as show in Table 3, it seems speculative decoding with smaller draft may achieve better speed up. Therefore, it seems smaller model such as draft model size can also work well on this task.**
>
> **A1:** We provide clarification on two key points:
>
> 1. For TPP data, large Transformer encoders (e.g., multi-layer, multi-head, with high-dimensional representations) are crucial for capturing complex temporal dependencies. Their larger parameter capacity allows them to model intricate data distributions and generate high-quality samples—albeit at the cost of slower autoregressive sampling.
>
> In contrast, smaller Transformer encoders (e.g., single-layer, single-head, with low-dimensional representations) are much faster at sampling due to reduced parameter capacity. However, they often struggle to model complex temporal patterns accurately, resulting in lower-quality samples that may not align well with the underlying data distribution [1,2,3].
>
> To summarize, while using a draft model for autoregressive sampling is indeed faster than using a target model, this speed advantage does not imply that the draft model alone is sufficient for high-quality modeling. Its limited expressiveness prevents it from capturing the true complexity of the data, which is why it is primarily used to assist rather than replace the target model.
>
> 2. Regarding the comment on “Notably, as show in Table 3......” we believe this is a misunderstanding. A smaller draft model indeed speeds up the speculative decoding process due to faster "draft" generation, but this can potentially lead to a lower acceptance rate. Nonetheless, our experiments show that, in general, smaller draft models do accelerate speculative decoding. However, as noted above, this does not imply that the draft model alone can achieve high-quality sampling, as its capacity is insufficient to model complex temporal patterns accurately.
>
> > **Q2: The proposed approach does not appear practical for real-world applications. Table 2 shows that the speed-up diminishes as K increases, and in real-world settings K is typically much larger. It is therefore unclear whether the method would remain effective in such scenarios.**
>
> **A2:** We respectfully disagree with this assessment. Our evaluation includes four established real-world datasets with substantial event type cardinalities ($K=16 \sim 22$), which are standard benchmarks in the TPP literature (Appendix B.2). These datasets represent realistic complexity encountered in practice. While we observe slight negative correlation between dataset cardinality $K$ and speed-up, the speed-up remains significant even for large $K$ (e.g., $5.849 \times$ speedup on Amazon with $K=17$, and $4.290\times$ speedup on Stackoverflow with $K=22$).
>
> > **Q3: Can you report the performance of small model, such as the draft model on those datasets?**
>
> **A3:** We evaluate the draft model $\mathcal{M}\_D$ (single-layer, single-head THP[1]) on both synthetic (Poisson, Hawkes, Multi-Hawkes) and real datasets (Taxi, StackOverflow, Amazon, Taobao), measuring sampling fidelity via $\Delta \mathcal{L}$ and $D\_{KS}$/$D\_{WS}$, and efficiency via wall-time.
>
> The simplistic architecture of $\mathcal{M}\_D$ struggles to capture temporal dependencies effectively even on synthetic datasets. Performance degrades further on real datasets, with significantly larger $\Delta \mathcal{L}$ and $D\_{WS}$ values, as $\mathcal{M}\_D$ cannot model the complex dependencies present in the data. Draft model alone performs poorly on this task, demonstrating the necessity of a large target model and the importance of our proposed TPP-SD framework, which leverages the speed of the draft model without sacrificing the fidelity provided by the target model.
>
> | Metric               | Sampling Type | Poisson | Hawkes | Multi-Hawkes |
> | -------------------- | ------------- | ------- | ------ | ------------ |
> | $\Delta \mathcal{L}$ | AR (draft)    | 2.174   | 1.348  | 1.305        |
> |                      | AR (target)   | 0.542   | 0.753  | 0.022        |
> |                      | TPP-SD        | 0.349   | 0.276  | 0.321        |
> | $D_{KS}$             | AR (draft)   | 0.153   | 0.141  | 0.126        |
> |                      | AR (target)   | 0.038   | 0.044  | 0.069        |
> |                      | TPP-SD        | 0.036   | 0.043  | 0.053        |
> | Wall-time            | AR (draft)    | 0.987   | 1.240  | 1.530        |
> |                      | AR (target)   | 3.477   | 5.147  | 4.007        |
> |                      | TPP-SD        | 1.647   | 2.547  | 1.893        |
>
> | Metric               | Sampling Type | Taxi  | StackOverflow | Amazon | Taobao |
> |:-------------------- |:------------- | -----:| -------------:| ------:| ------:|
> | $\Delta \mathcal{L}$ | AR (draft)    | 1.504 | 1.136         | 0.614  | 2.122  |
> |                      | AR (target)   | 0.441 | 0.587         | 0.056  | 0.446  |
> |                      | TPP-SD        | 0.065 | 0.231         | 0.129  | 0.033  |
> | $D^t_{WS}$           | AR (draft)    | 0.765 | 0.994         | 0.534  | 0.577  |
> |                      | AR (target)   | 0.201 | 0.470         | 0.189  | 0.236  |
> |                      | TPP-SD        | 0.082 | 0.391         | 0.078  | 0.076  |
> | $D^k_{WS}$           | AR (draft)    | 0.768 | 0.821         | 0.759  | 1.384  |
> |                      | AR (target)   | 0.055 | 0.376         | 0.184  | 0.267  |
> |                      | TPP-SD        | 0.655 | 0.375         | 0.418  | 0.751  |
> | Wall-time            | AR (draft)    | 0.150 | 0.120         | 0.070  | 0.410  |
> |                      | AR (target)   | 1.157 | 1.353         | 1.023  | 5.890  |
> |                      | TPP-SD        | 0.453 | 0.700         | 0.290  | 3.460  |
>
> References:
>
> [1] Transformer Hawkes Process, ICML, 2020.
>
> [2] Self-Attentive Hawkes Process, ICML, 2020.
>
> [3] Transformer Embeddings of Irregularly Spaced Events and Their Participants, ICLR, 2022.

---

> ### Author Response · Authors · 2025-08-04
> **Kind reminder to Reviewer HqH3**
>
> Dear Reviewer HqH3,
>
> Thank you again for your evaluation of our work and the valuable feedback you have provided. We have posted a detailed rebuttal to address the concerns you raised. As the author-reviewer discussion phase will conclude in less than two days, we kindly request your feedback and are keen to know if our clarifications and additional results have effectively addressed your questions.
>
> If any points remain unclear, we are happy to engage in further discussion. We sincerely hope that our rebuttal clarifies the merits of our work, and we would appreciate it if you would consider our response in your final evaluation and revisit your rating.
>
> Best regards,
>
> Authors of Submission 6980

---

> ### Author Response · Authors · 2025-08-08
>
> Dear Reviewer HqH3,
>
> Just a quick reminder that the discussion period closes in **one** day.
>
> We believe we have **fully addressed all of your concerns** and have also **provided additional experimental results**. If you have any further comments or remaining concerns, we would greatly appreciate hearing them before the discussion period ends.
>
> Best regards,
>
> The Authors

---

### Decision · Program_Chairs · 2025-09-17

**Decision:**

Accept (poster)

**Comment:**

This paper integrates speculative decoding techniques into transformer-based temporal point-process sampling, which achieves 2 to 6 times speed-up on both synthetic and real datasets. All the reviewers appreciated the methodological contributions in the paper, although the idea might not be utterly novel.

Several concerns are lacking baselines and insufficient discussion in the related work on AR sampling. During the rebuttal, authors provided additional numerical results to lift these concerns. Although formally incorporating these results require some extra work, it is believed that these results can be added within a short period. Therefore, I recommend acceptance of the paper.